# Identification of the FSH-RH as the other gonadotropin-releasing hormone

Shun Kenny Uehara [1,6], Yuji Nishiike [2,6], Kazuki Maeda[1], Tomomi Karigo [3], Shigehiro Kuraku [4,5], Kataaki Okubo [2] & Shinji Kanda [1]✉

In vertebrates, folliculogenesis and ovulation are regulated by two distinct pituitary gonadotropins: follicle-stimulating hormone (FSH) and luteinizing hormone (LH). Currently, there is an intriguing consensus that a single hypothalamic neurohormone, gonadotropin-releasing hormone (GnRH), regulates the secretion of both FSH and LH, although the required timing and functions of FSH and LH are different. However, recent studies in many non-mammalian vertebrates indicated that GnRH is dispensable for FSH function. Here, by using medaka as a model teleost, we successfully identify cholecystokinin as the other gonadotropin regulator, FSH-releasing hormone (FSH-RH). Our histological and in vitro analyses demonstrate that hypothalamic cholecystokinin-expressing neurons directly affect FSH cells through the cholecystokinin receptor, Cck2rb, thereby increasing the expression and release of FSH. Remarkably, the knockout of this pathway minimizes FSH expression and results in a failure of folliculogenesis. Here, we propose the existence of the "dual GnRH model" in vertebrates that utilize both FSH-RH and LH-RH.

The pituitary gland secretes two essential gonadotropin hormones for gonadal functions: follicle-stimulating hormone (FSH) and luteinizing hormone (LH). Due to the importance of sexual development and reproduction in all vertebrates studied, the regulatory mechanism of these hormones has been extensively investigated. Importantly, given the hypothesis that the secretion of anterior pituitary hormones is mostly regulated by the hypothalamic neuropeptides[1,2], LH-releasing hormone (LH-RH) was discovered in 1971 as the key regulator of LH and FSH release in mammals, marking a significant breakthrough[1,3,4]. LH-RH was later studied in other mammals, and its involvement in reproductive regulation has been generally established in vertebrates[5]. Consequently, it has been widely recognized that LH-RH is the sole gonadotropin-releasing hormone (GnRH), responsible for regulating both FSH and LH secretion in vertebrates[6], and it is now referred to as

the final common pathway for central regulation of gonadotropin secretion or fertility[7–9].

However, this once-established consensus has been challenged in vertebrates other than mammals. Intriguingly, it has been reported that GnRH knockout (KO) does not affect FSH function in model teleosts such as medaka and zebrafish[10,11], which implies that GnRH may not be the primary regulator of FSH release, at least in teleosts. Also, in quail, a model avian species, reports clearly state that the effects of GnRH on FSH are smaller than those on LH release[12,13]. This observation raises the same perplexing mystery indicated in teleosts in that GnRH has a small effect on FSH[1,14–16]. Thus, it is postulated that an exclusive FSH-regulating hormone, FSH-RH, exists in vertebrates alongside GnRH. This hypothesis suggests that a significant number of vertebrates, rather than being limited to minor subgroups, may utilize

[1]Atmosphere and Ocean Research Institute, The University of Tokyo, Chiba, Japan. [2]Department of Aquatic Bioscience, Graduate School of Agricultural and Life Sciences, The University of Tokyo, Tokyo, Japan. [3]Kennedy Krieger Institute, Solomon H. Snyder Department of Neuroscience, Johns Hopkins University School of Medicine, Baltimore, MD, USA. [4]Laboratory for Phyloinformatics, RIKEN Center for Biosystems Dynamics Research, Kobe, Japan. [5]Present address: Molecular Life History Laboratory, Department of Genomics and Evolutionary Biology, National Institute of Genetics, Mishima, Japan. [6]These authors contributed equally: Shun Kenny Uehara, Yuji Nishiike. ✉e-mail: shinji@aori.u-tokyo.ac.jp

distinct FSH-RH and LH-RH systems, indicating the presence of a "dual GnRH model."

In the present study, to reevaluate the current "solo GnRH model," we aimed to identify the FSH-RH. Among non-mammalian vertebrates, we used medaka because of their amenability to genetic modification and the exceptionally accessible information about their reproductive biology[9,17–20]. The use of medaka is also advantageous because, as teleosts, they have separate cells that express FSH or LH, unlike mammals, in which FSH and LH are secreted from a single cell type, making medaka useful for examination of the FSH-RH receptors[21–25].

In this work, we successfully identified cholecystokinin (CCK) expressed in the hypothalamus as the FSH-RH. First, FSH cell-specific RNA-seq demonstrates that cholecystokinin 2 receptor b (*cck2rb*) is highly expressed in FSH cells. Intriguingly, disruption of this receptor gene drastically reduces the expression of *FSH subunit beta* gene (*fshb*) expression in the pituitary, resulting in the failure of folliculogenesis. Additionally, this severe phenotype is rescued by a rescue transgene that induces expression of Cck2rb specifically in FSH cells, which demonstrates that the Cck2rb expressed in FSH cells is exclusively important for FSH function.

We also demonstrate in in vitro experiments that CCK peptide drastically increases $[Ca^{2+}]_i$ of FSH cells as well as *fshb* expression in the pituitary, which suggests its significant role in the upregulation of both release and expression of FSH. CCK is also shown to be expressed in pituitary-projecting neurons in the hypothalamus, indicating its role as a hypophysiotropic factor. Finally, KO of *cck* genes results in the drastic attenuation of FSH function. Based on these lines of evidence, we hypothesize that CCK is the primary regulator of FSH secretion, namely FSH-RH.

## Results

### Cck2rb is highly expressed in FSH cells
First, considering the importance of identifying the receptor expressed in the FSH cells, we performed FSH-cell specific RNA-sequencing (RNA-seq) by collecting green fluorescent protein (GFP)-positive cells in the pituitary of FSH-GFP medaka ($n = 4$, Supplementary Fig. 1). Among the metabotropic receptors expressed, we found that *cck2rb* had the highest expression (Supplementary Table 1; a phylogenetic tree of Cck receptors is shown in Supplementary Fig. 2). Therefore, we conducted double in situ hybridization of the *fshb* or *LH subunit beta* gene (*lhb*) and *cck2rb* to examine their co-expression in the pituitary through the histological method. From these experiments, we demonstrated that *cck2rb* is expressed exclusively in FSH cells (Fig. 1a, b).

### Cck2rb is essential for FSH function
To determine the functional essentiality of the identified receptor, we generated *cck2rb* KO and analyzed phenotypes in both females and males. Our analysis revealed that both females and males of *cck2rb*−/− have substantially smaller gonads compared to those of *cck2rb*+/+ or *cck2rb*+/− individuals (Fig. 1c). In addition, a severe reduction in gonadal weight (normalized by body weight, gonadosomatic index/GSI), was observed in both *cck2rb*−/− females and males (*cck2rb*+/+ and *cck2rb*+/− $n = 5$, *cck2rb*−/− $n = 4$, Fig. 1d; $n = 5$, Fig. 1e), whereas there was no apparent change in body weight (Supplementary Fig. 3). No spawning was observed in *cck2rb*−/− while fertilized eggs were observed in *cck2rb*+/+ and *cck2rb*+/− fish.

To clarify whether male or female has dysfunction in spawning, *cck2rb*−/− male or female was paired with a wild type partner and the number of eggs spawned and rate of fertilization were observed. The eggs spawned and the fertilization rate in males were similar in *cck2rb*−/− or *cck2rb*+/+, while no spawning was observed from *cck2rb*−/− females paired with wild type males ($n = 4$, Supplementary Fig. 4a–c). These results indicate that the dysfunction of spawning is owing to *cck2rb*−/− females. To detail this dysfunction of females, the number of eggs

spawned by *cck2rb*+/+, *cck2rb*+/−, and *cck2rb*−/− females were paired with wild type males of each genotype during 60 days post-hatch (dph) to 105 dph was counted. It was shown that *cck2rb*+/+ and *cck2rb*+/− started to spawn eggs around 70 dph and the number of eggs peaked around 90 dph, whereas *cck2rb*−/− did not spawn any eggs by 105 dph (Supplementary Fig. 4d, e).

Additionally, histological analysis revealed that the ovaries of the *cck2rb*−/− females include only the previtellogenic oocyte, whereas wild type ovaries show full-grown oocytes (Fig. 1f). Unlike females, *cck2rb*−/− males showed normal spermatogenesis (Fig. 1f), which is similar to the results of *fshb* KO medaka, in that only females show complete infertility[26,27]. Estradiol (E2) content of the whole body of the female ($n = 3$) was analyzed in *cck2rb*+/+ and *cck2rb*−/− fish and indicated that the E2 concentration was significantly lower in *cck2rb*−/− than in *cck2rb*+/+ fish (Fig. 1g).

Finally, the pituitaries of *cck2rb*+/− and *cck2rb*−/− fish ($n = 7$) were subjected to quantitative reverse transcription polymerase chain reaction (qRT-PCR) to examine the effects of KO on gonadotropin gene expression in females and males (Fig. 1h–m). In both sexes, *fshb* expression was much lower in *cck2rb*−/− than in *cck2rb*+/− (Fig. 1h, k). Since FSH is essential for folliculogenesis[26] in females, this reduction of *fshb* is the reason for the small gonads and low E2 concentration observed in *cck2rb*−/− (Fig. 1g).

Although there are no previous studies that have examined the GSI of *fshb* KO males in medaka, FSH is generally considered to stimulate spermatogenesis including spermatogonial proliferation in teleosts[27–29]. Therefore, the phenotype of *cck2rb*−/− males that show reduced GSI (Fig.1c) can also be explained by their reduction in FSH secretion (Fig. 1k). Thus, Cck2rb was shown to have a pivotal role in FSH secretion in both sexes.

Interestingly, only in females, the expression of the *lhb* was lower in *cck2rb*−/− than in *cck2rb*+/− (Fig. 1i, l). Because the *lhb* expression is upregulated by the ovarian estrogen[24], we considered that the decrease of *lhb* might be the secondary effect of the drastic decline in the serum estrogen in *cck2rb*−/− (Fig. 1g). To diminish this secondary effect, we compared the expression after ovariectomy. In the ovariectomized females ($n = 6$), there was no difference in *lhb* expression between *cck2rb*+/− and *cck2rb*−/− (Supplementary Fig. 5). These results strongly suggest that the difference in *lhb* expression between intact *cck2rb*+/− and *cck2rb*−/− females should be due to the secondary effect of reduced serum estrogen concentration in *cck2rb*−/−. Expression of *thyroid stimulating hormone subunit beta* gene (*tshb*) did not change in either sex (Fig. 1j, m).

### FSH cell-specific Cck2rb rescue recovers the FSH function
The essentiality of Cck2rb in FSH cells was further examined by rescuing *cck2rb* specifically in FSH cells in *cck2rb*−/− medaka. Here, we generated a transgenic medaka harboring a rescue transgene containing C-terminal FLAG-tagged *cck2rb* coding sequence under the promoter of *fshb*, which expresses Cck2rb specifically in FSH cells (Fig. 2a). After crossing with *cck2rb* KO medaka, the effect of the rescue transgene was examined. *cck2rb*−/− medaka with the rescue transgene showed spawning unlike their siblings without the transgene. To further assess the effect of the rescue transgene, the ovary and testis sizes of *cck2rb*−/− medaka with and without the rescue transgene were examined. Our findings revealed a significant increase in gonadal size for both the ovary ($n = 6$) and testis ($n = 4$) when the transgene was present, compared to their siblings without the transgene, in both sexes (Fig. 2b, f).

We also analyzed the mRNA expression of *fshb*, *lhb*, and *tshb* in the pituitary using qRT-PCR. Consistent with the ovary and testis sizes, the pituitary with the rescue transgene showed a significant increase in the *fshb* mRNA expression compared to those without the transgene, in both females and males (Fig. 2c, g). In females, the *lhb* expression of the rescued KO showed a significant increase, which can be explained

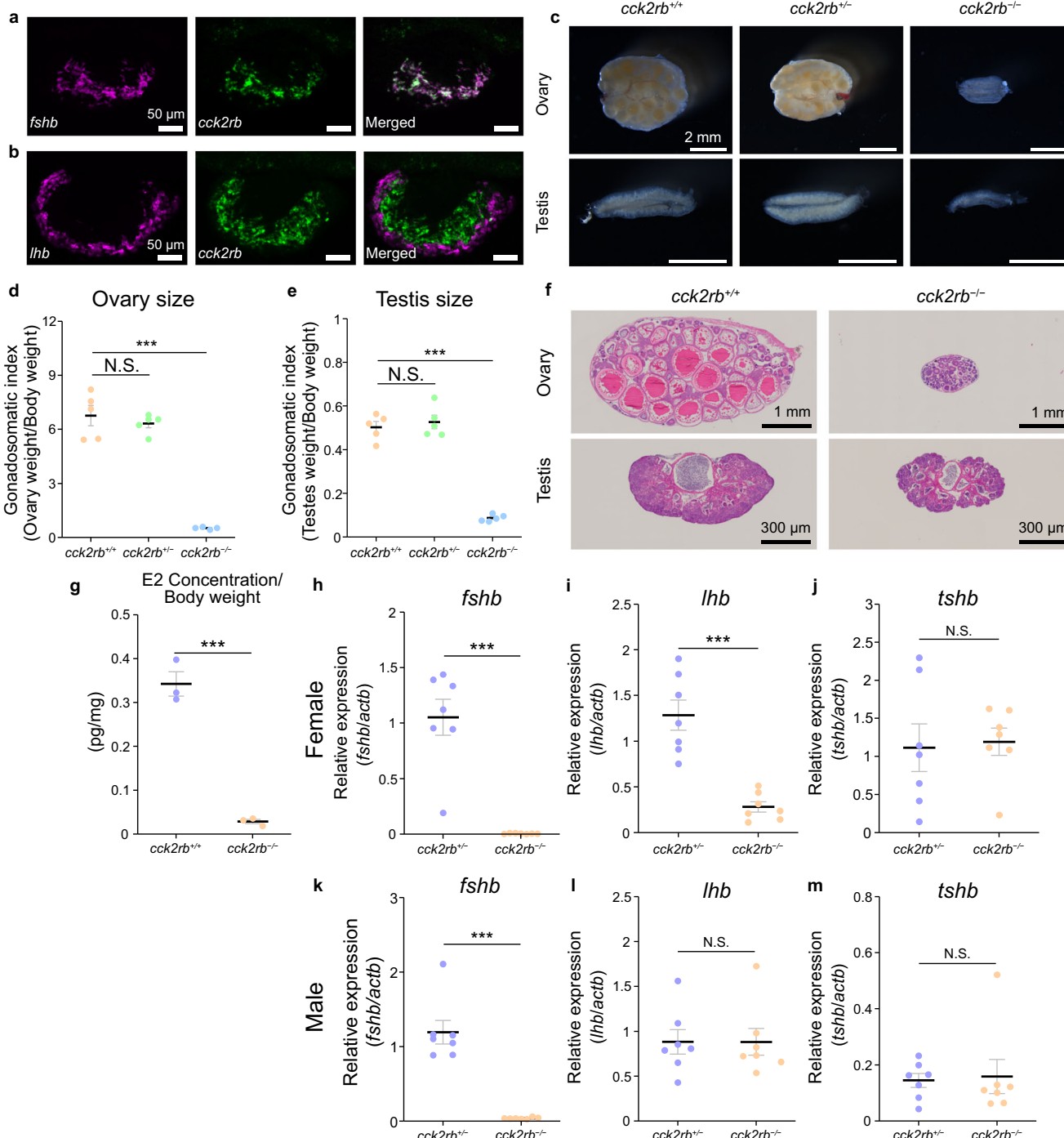

**Fig. 1 | Cck2rb expressed in the pituitary FSH cells is essential for normal FSH expression. a, b** Double in situ hybridization of the pituitary of medaka labeled with *fshb* (**a**), *lhb* (**b**), and *cck2rb* (*n* = 4 fish). **c–e** Gross morphology of gonads (**c**) as well as ovarian size (**d**) and testis size (**e**) of *cck2rb*+/+ and *cck2rb*−/− females and males (*n* = 5 fish; ***, *p* = 2.3e-6, 1.9e-6). **f** Histological sections of the ovary and testis of *cck2rb*+/+ and *cck2rb*−/−. **g** The estradiol (E2) content of females was significantly lower than that of wild type medaka (*n* = 3 fish; *p* = 0.0038). **h–m** Expression of *fshb* (**h, k**), *lhb* (**i, l**), and *tshb* (**j, m**) in the pituitary of each genotype (*n* = 7 fish; ***, *p* = 3.2e-5, 9.2e-5, 9.8e-6). The data are mean ± SEM. ****p* < 0.001, N.S. not significant, two-sided Dunnett's test (**d–e**), two-sided Student's *t* test (**g–m**). Source data are provided as a Source Data file.

by the secondary effect of the rescued FSH function (Fig. 2d). In contrast, in males, no significant change in *lhb* expression was observed (Fig. 2h), which is consistent with the KO result that showed no reduction in *lhb* expression in males (Fig. 1l). The *tshb* expression did not change in either group, both in females and males (Fig. 2e, i). Thus, Cck2rb, which is expressed in FSH cells, has been proven to be crucial for FSH regulation.

## CCK is expressed in the hypophysiotropic neurons

After identifying Cck2rb as the prime candidate for the FSH-RH receptor, the ligands for Cck2rb were explored. In vertebrates, genes for ligands that bind to Cck2rb are members of the Gastrin/Cholecystokinin family; medaka have two CCK paralogs: *cholecystokinin a* (*ccka*) and *cholecystokinin b* (*cckb*), as well as *gastrin (gast)*[30] (Phylogenetic tree, Supplementary Fig. 6; sequence alignment,

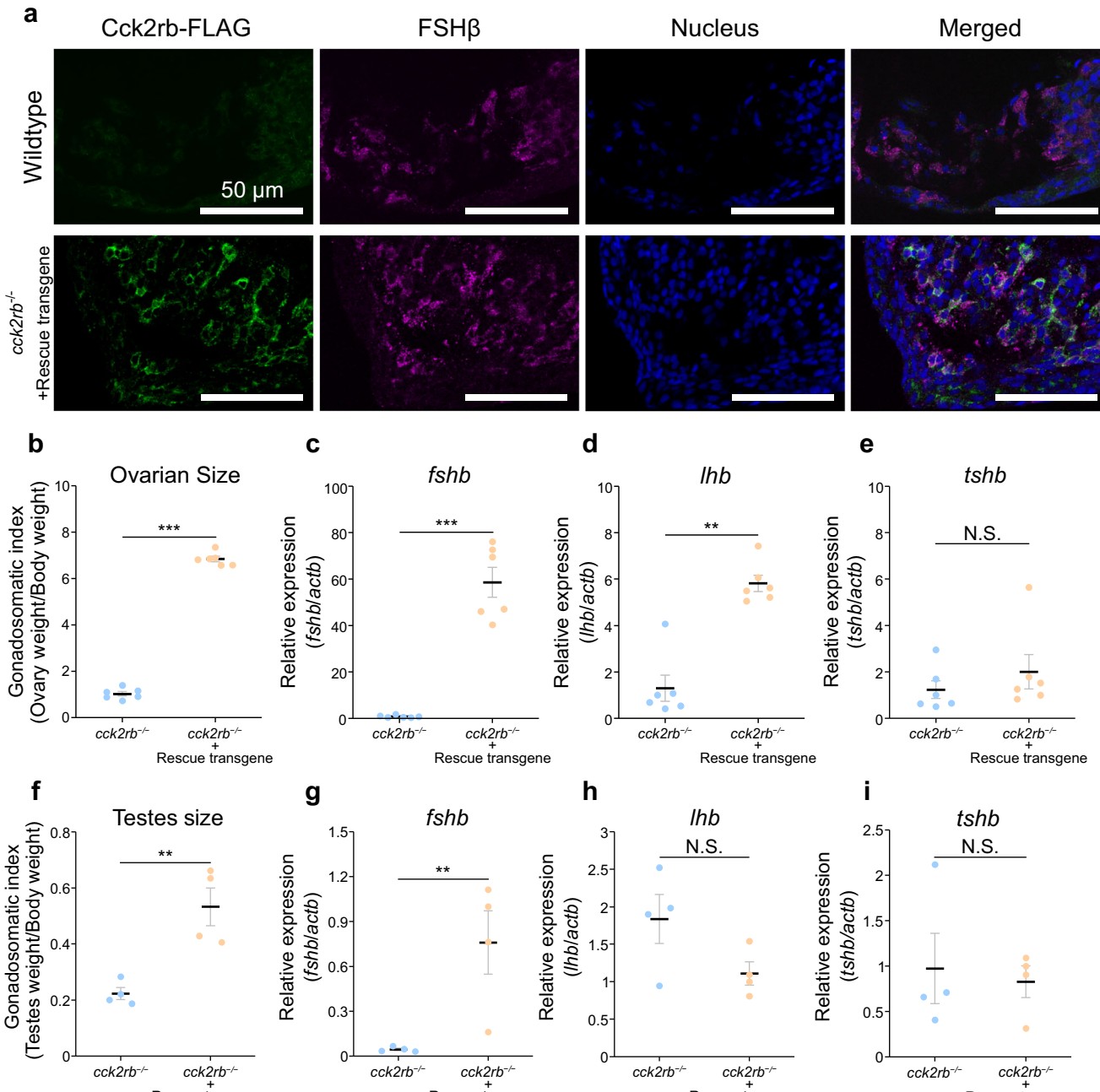

**Fig. 2 | FSH cell-specific rescue of Cck2rb in *cck2rb* KO medaka recovered *fshb* expression and fertility in females. a** Immunohistochemistry of the pituitary of wild type and *cck2rb*<sup>−/−</sup> medaka with the rescue transgene (*fshb*:Cck2rb-FLAG). The pituitary is labeled with transgenically introduced Cck2rb (FLAG-tagged, green) and intrinsic FSHβ (magenta). FSH cells are successfully forced to express FLAG-tagged Cck2rb. Note that the pituitary of wild type medaka shows normal expression of FSH but shows no immunoreactivity to FLAG. **b** Ovarian size of *cck2rb*<sup>−/−</sup> females with or without the rescue transgene (*n* = 6 fish; *p* = 3.9e-7). The data are mean ± SEM. **c**–**e** Expression of *fshb* (**c**), *lhb* (**d**), and *tshb* (**e**) in the pituitary of each *cck2rb*<sup>−/−</sup> with or without the rescue transgene (*n* = 6 fish; ****p* = 0.00028; ***p* = 0.0010). **f** Testes size of *cck2rb*<sup>−/−</sup> males with or without the rescue transgene (*n* = 4 fish; ****p* = 0.0046). The data are mean ± SEM. **g**–**i** Expression of *fshb* (**g**), *lhb* (**h**), and *tshb* (**i**) in the pituitary of each *cck2rb*<sup>−/−</sup> with or without the rescue transgene (*n* = 4 fish; ***p* = 0.0022). The data are mean ± SEM. The data are mean ± SEM. ****p* < 0.001, ***p* < 0.01, **p* < 0.05, N.S. not significant, two-sided Student's *t* test (**b**–**i**). Source data are provided as a Source Data file.

Supplementary Fig. 7). Because the post-transitionally cleaved and sulfated forms of 8-amino acid peptide of Ccka and Cckb (CCK-8s: DY(SO₃H)LGWMDF-$_{NH2}$ in medaka) and Gastrin (Gastrin-8s: DY(SO₃H)RGWLDF-$_{NH2}$ in medaka) are reported to show sufficient biological activity in various species, we used them for the luciferase reporter assay. Note that both *ccka* and *cckb* genes contain the identical deduced 8-amino acid residues corresponding to CCK-8s.

In the luciferase assay, for all cyclic adenosine monophosphate (cAMP), Ca²⁺, and mitogen-activated protein kinase (MAPK) reporter

systems, Cck2rb was activated by both CCK-8s and Gastrin-8s in dose-dependent manners (Fig. 3a). The half-maximal effective concentrations (EC₅₀) are summarized in Supplementary Table 2. Thus, CCK and Gastrin can activate the receptor, Cck2rb, at physiological concentrations.

The ligands were further analyzed for their expression in the hypothalamus. Because reverse transcription PCR (RT-PCR) of the whole brain indicated high expression of *ccka* and *cckb*, whereas *gast* was not detected in the PCR with the same cycles

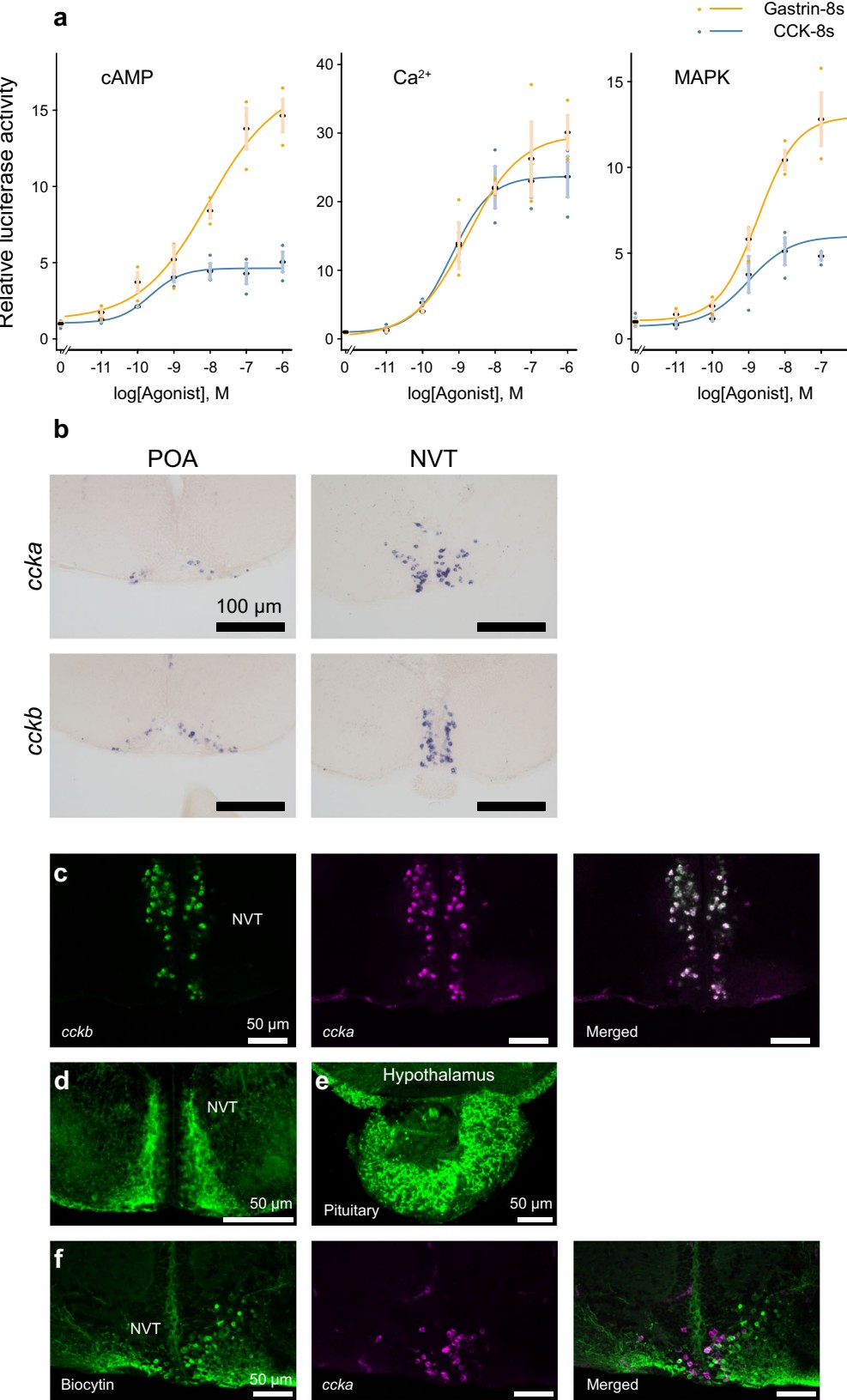

**Fig. 3 | Hypophysiotropic neurons expressing *ccka* and *cckb* exist in the hypothalamus. a** Luciferase assay of cholecystokinin family peptides CCK-8s and Gastrin-8s for cAMP, Ca²⁺, and MAPK pathways in HeLa cells expressing Cck2rb ($n = 3$ individual wells). The data are mean ± SEM. **b** In situ hybridization of *ccka* and *cckb* mRNA in the preoptic area (POA) and the nucleus ventralis tuberis (NVT), of the hypothalamus. **c** Double in situ hybridization of the *ccka* (magenta) and *cckb* (green) in the NVT ($n = 4$ fish). **d**, **e** Immunohistochemistry (IHC), using a CCK-specific antibody labeling the CCK neurons in the NVT (**d**) and axonal fibers in the pituitary (**e**) ($n = 2$ fish). **f** Dual labeling of retrogradely introduced biocytin (green) from the pituitary and *ccka* mRNA (magenta) ($n = 3$ fish). Source data are provided as a Source Data file.

(Supplementary Fig. 8), we examined the localization of their mRNAs in the brain by in situ hybridization, using *ccka* and *cckb* probes to analyze their possible expression. Here, *ccka* and *cckb* were expressed in various regions of the brain including the nuclei that contain the hypophysiotropic (pituitary-projecting) neurons, the preoptic area (POA), and the nucleus ventralis tuberis (NVT) in the hypothalamus (Fig. 3b). Because the expression of *ccka* and *cckb* in the NVT apparently overlapped, we examined the possible co-expression of *ccka* and *cckb* by double in situ hybridization. We discovered that many CCK neurons in NVT co-express *ccka* and *cckb* mRNA (Fig. 3c).

To observe their projection, we used a commercial cholecystokinin antibody. As expected from previous studies of comparative anatomy[31,32], cell bodies in the nuclei including the hypothalamus (Fig. 3d), as well as dense axonal fibers in the pituitary (Fig. 3e), were labeled. Note that all cell bodies and fibers were diminished in the double knockout of *ccka* and *cckb*, which strongly suggests the specificity of this immunohistochemistry (Supplementary Fig. 9). Therefore, the cell bodies and axonal projection in the pituitary were suggested to be originated from *ccka*- and (or) *cckb*-expressing neurons. Next, to examine which population of these neurons projected to the pituitary, we performed retrograde tracer labeling using biocytin from the pituitary in combination with *ccka* in situ hybridization. In the sample that underwent biocytin injection in the area where FSH cells are localized in the pituitary, we observed co-labeling of retrogradely labeled biocytin and *ccka* mRNA in the neurons in the NVT (Fig. 3f). These results strongly suggest that the CCK neurons in the NVT are hypophysiotropic and project their axons to the pituitary FSH cells.

### CCK increases intracellular Ca²⁺ of FSH cells

To examine the possibility that the release of FSH is induced by CCK, $Ca^{2+}$ imaging of FSH cells was performed. The pituitaries of females were isolated from the *fsh*:inverse pericam (*fsh*:IP) transgenic medaka[33], whose FSH cells express a genetically encoded $Ca^{2+}$ indicator called inverse pericam[34]. In FSH cells, 1 µM CCK-8s induced a rapid and strong intracellular $Ca^{2+}$ increase, which triggers hormonal release from the FSH cells (Fig. 4a). This effect was also observed in males (Supplementary Fig. 10). Perfusion of various concentrations of CCK-8s ($n = 4$) demonstrated that this action on FSH cells was dose-dependent ($EC_{50} = 19$ nM, Fig. 4b). Unlike that in FSH cells, the $Ca^{2+}$ imaging experiment in LH cells using the *lh*:IP transgenic medaka[33] ($n = 4$) did not show a CCK-8s-induced $Ca^{2+}$ response (Fig. 4c, d).

Additionally, we examined the $Ca^{2+}$ response when perfused with Gnrh1 (mdGnRH) peptide[35], which is considered to be the intrinsic GnRH subtype controlling gonadotropin secretion[26,36–38]. Although mdGnRH (100 nM) induced a $Ca^{2+}$ response in FSH cells as suggested in a previous study[33], this response was significantly lower than that induced by CCK-8s at the same concentration ($n = 5$, Supplementary Fig. 11a, b). Furthermore, mdGnRH showed a large $Ca^{2+}$ response in LH cells while CCK-8s showed no effect on LH cells ($n = 5$, Supplementary Fig. 11c, d). Therefore, we suggest here that CCK can be the primary hypophysiotropic factor to regulate FSH release, while mdGnRH is the primary factor to regulate LH release. Note that this conclusion applies regardless of sex, as evidenced in the present study (Fig. 4a and Supplementary Fig. 10) as well as in a previous study[33].

### CCK increases *fshb* expression in vitro

We next examined if CCK stimulates FSH expression by an in vitro experiment with isolated pituitaries. The pituitaries of female medaka ($n = 5$) were isolated and incubated for 48 h in a culture medium containing 0, 10, or 100 nM CCK-8s. After incubation, the expression of *fshb*, *lhb*, and *tshb* mRNA was analyzed by qRT-PCR. It was revealed that *fshb* expression was dose-dependently increased by CCK-8s (Fig. 4e–g). In the presence of 10 or 100 nM CCK-8s during incubation,

*fshb* increased ~10-fold compared to the 0 nM control (Fig. 4e). Although the expression of *lhb* mRNA showed an apparent increase, it was not statistically significant (Fig. 4f). CCK-8s did not affect *tshb* expression (Fig. 4g). Also, a similar experiment incubating the pituitary in 0 nM and 100 nM CCK-8s ($n = 5$ for females, $n = 4$ for males) showed similar results in both sexes (Supplementary Fig. 12).

Although we see a tendency for an increase in *lhb* expression as well, considering the results of $Ca^{2+}$ imaging and double in situ hybridization, it is highly probable that LH cells do not possess CCK receptors. It is therefore reasonable that we did not observe a significant increase in *lhb* expression.

The same procedure was conducted using the pituitaries of *cck2rb*⁺/⁻ and *cck2rb*⁻/⁻ medaka. As expected, CCK-8s increased *fshb* expression in the *cck2rb*⁺/⁻ pituitary but not in the *cck2rb*⁻/⁻ pituitary (Fig. 4h). For other glycoprotein hormone genes, similar to the results of the wild type medaka, CCK-8s did not significantly increase the expression of *lhb* or *tshb* in both *cck2rb*⁺/⁻ and *cck2rb*⁻/⁻ genotypes (Fig. 4i, j). These results indicate that CCK increases the expression of *fshb* via Cck2rb at the pituitary level.

### CCK is essential for FSH function

From the results above, we showed that CCK strongly stimulated FSH secretion, in both hormone synthesis and release. To examine the necessity of intrinsic CCK, we generated KOs of the two CCK paralogs, *ccka* and *cckb*, by CRISPR[39,40], and observed the phenotype in ~3 months after hatch. Interestingly, although a single KO of *ccka* or *cckb* resulted in a normal phenotype, the double KOs showed a severe change in phenotype (Fig. 5a, Supplementary Fig. 13).

The overall phenotype of the double KO was similar to that of the *cck2rb* KO. In both females and males, the gonadal size of *ccka*/*cckb* double KO was drastically decreased (Fig. 5a–c and Supplementary Fig. 13). As expected, pituitary mRNA expression ($n = 6$) of *fshb* in *ccka*/*cckb* double KO was much lower than that of the other genotypes (Fig. 5d). Also, *lhb* expression decreased in the double KO, whereas *tshb* did not differ among all genotypes, which is the same situation as *cck2rb* KO (Fig. 5e, f). The fact that only the double KO was associated with a severe phenotype suggests that *ccka* and *cckb* have redundant functions in FSH regulation, which is also consistent with the fact that *ccka* and *cckb* co-localize in the NVT hypophysiotropic neurons (Fig. 3c).

To examine the fertility, *ccka*/*cckb* KO male or female was paired with a hetero partner and the number of eggs spawned and rate of fertilization were observed. As expected, successful spawning was observed in *ccka*/*cckb* double KO males, while no spawning was observed in *ccka*/*cckb* double KO females ($n = 4$, Supplementary Fig. 14a–c). These results indicate that the dysfunction of spawning is the result of *ccka*/*cckb* KO females, which is consistent with the results of receptor KOs.

Interestingly, some of the double knockouts started spawning about one month after the wild type started to spawn (Supplementary Fig. 14d, e) even with completely reduced *fshb* expression and significantly smaller ovary (n = 4, Supplementary Fig. 15). This can be explained by the compensatory mechanisms involving LH as follows: First, a small amount of FSH (with *fshb* expression of <7% of wild type) is released by another ligand of Cck2rb, perhaps Gastrin, which slowly stimulates folliculogenesis. This causes an increase in serum estrogen, which should induce LH secretion in a positive feedback manner[24]. Here, in medaka, because LH can also activate FSH receptor[41], LH can regulate the ovulatory cycle in an FSH-independent manner once LH secretion is activated.

Despite the occurrence of this delayed spawning, the overall results consistently indicate a severe deficiency of the FSH system in *ccka*/*cckb* double KO. Therefore, cholecystokinin is the primary factor responsible for the expression and release of FSH, which should be referred to as the FSH-RH.

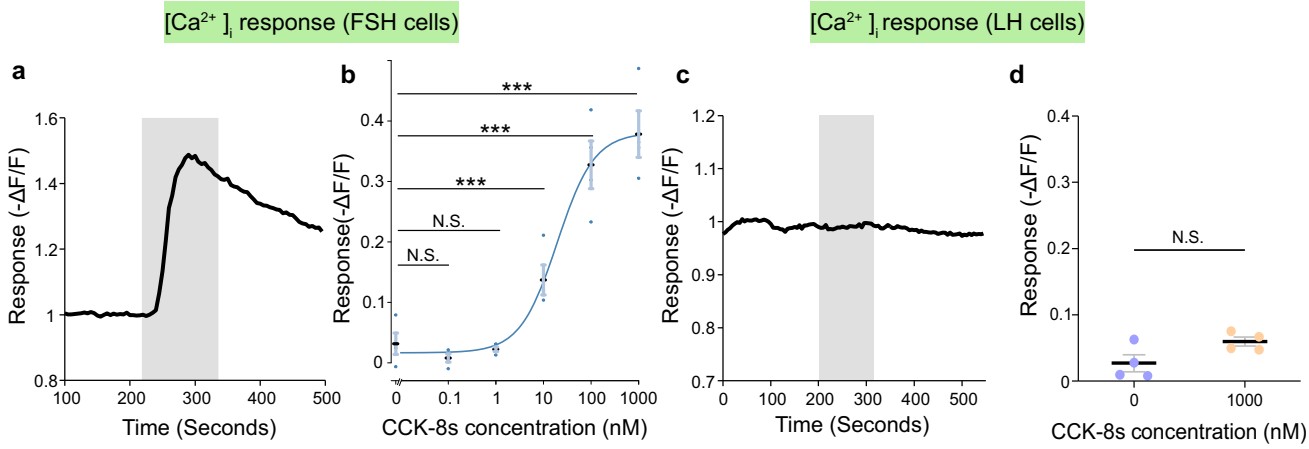

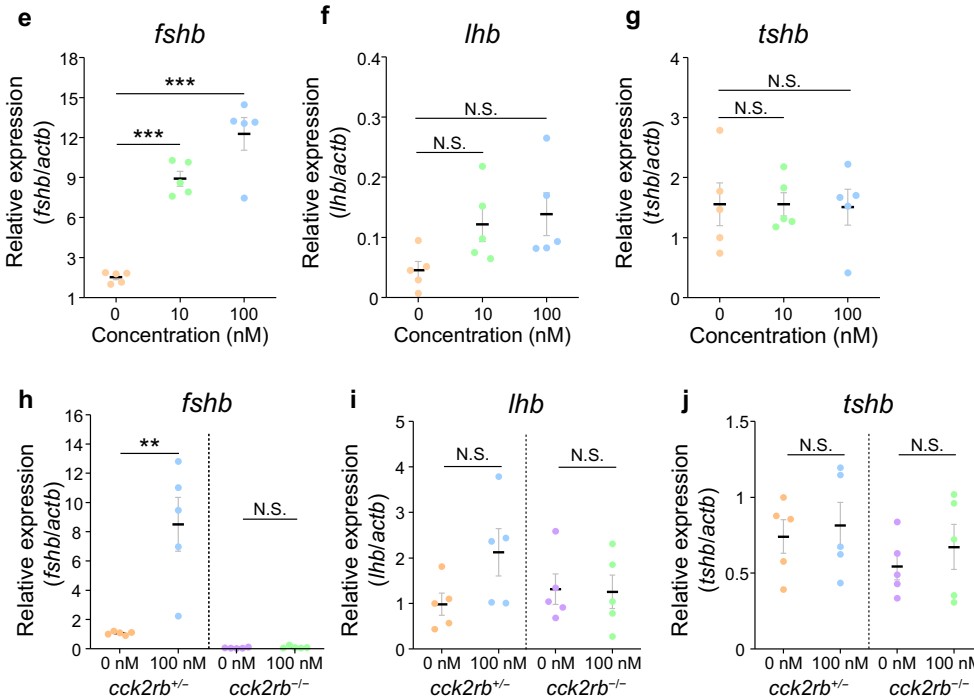

**Fig. 4 | CCK robustly increased both [Ca²⁺]ᵢ, which causes the hormonal release, and mRNA expression of *fshb*. a** Ca²⁺ imaging of FSH cells applied with 1 µM CCK-8s. The shaded section indicates the time and duration of the perfusion of CCK-8s. **b** Dose-response curve of [Ca²⁺]ᵢ in FSH cells applied with CCK-8s of various concentrations ($n = 4$ fish;***$p = 1.4e-6$, $1.4e-6$, $2.2e-4$). **c** Ca²⁺ imaging of LH cells applied with 1 µM CCK-8s. The shaded section indicates the time and duration of the perfusion of CCK-8s. **d** Repetitive Ca²⁺ imaging trials on LH cells ($n = 4$ fish). **e–g** qRT-PCR of the pituitary after incubating with CCK-8s for 48 h (**e**, *fshb*; **f**, *lhb*; **g**, *tshb*; $n = 5$ fish; ***$p = 2.9e-6$, $4.8e-5$). **h–j** qRT-PCR of the pituitary of *cck2rb⁺/⁻* and *cck2rb⁻/⁻* after incubating in CCK-8s (**h**, *fshb*; **i**, *lhb*; **j**, *tshb*; $n = 5$ fish; **, $p = 3.7e-3$). The data are mean ± SEM. ***$p < 0.001$, **$p < 0.01$, N.S. not significant, two-sided Dunnett's test (**b**, **e–g**), two-sided Student's *t* test (**d**, **h–j**). Source data are provided as a Source Data file.

## Discussion

In contrast to the current "solo GnRH model" hypothesis established in mammals, we identified hypothalamic CCK as the most potent and essential regulator of FSH release. The strong effects shown in in vitro experiments and the severe phenotype in knockout experiments strongly suggest that CCK is the FSH-RH. It was surprising that the long-absent FSH-RH was proven to be CCK, which is well-known as an intestinal peptide whose function has been extensively studied in the mammalian digestive system in both basic and clinical research contexts[42].

To date, despite many trials mainly in rodent models, a strong regulator of FSH other than GnRH has not been identified, thus it was

hypothesized that LH-RH/GnRH is the only gonadotropin-releasing hormone in vertebrates. However, our discovery of FSH-RH provides strong counterevidence of this hypothesis. The evidence provided in this study suggests the existence of species in which FSH and LH are regulated by distinct hypothalamic factors, namely the "dual GnRH model" (Fig. 6a).

Although this study provides evidence suggesting that CCK acts as the FSH-RH in medaka, there are several indications that this mechanism could extend to other vertebrates. First, its mechanisms are likely to be conserved at least in all teleosts. A cell type-specific RNA-seq study indicated that FSH-expressing cells showed high expression of CCK receptors in another teleost, tilapia[43,44]. More

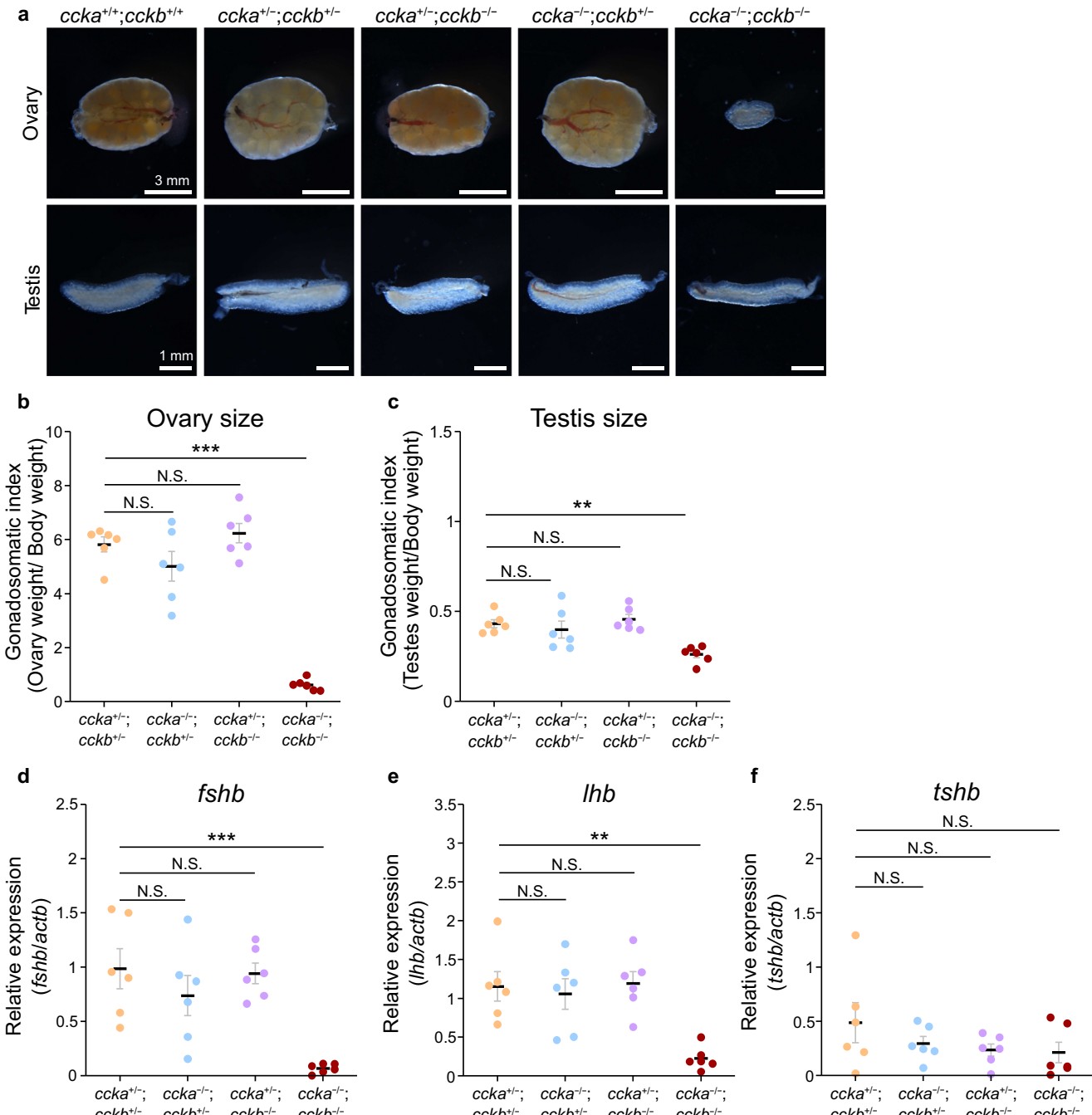

**Fig. 5 | Ccka and Cckb redundantly function as FSH-RH. a** Gross morphology of the ovaries and testes of *ccka* and *cckb* single and double KO. **b, c** The ovarian size (**b**) and testis size (**c**) of each genotype normalized by body weight (*n* = 6 fish; ***p = 1.4e-6; **p = 0.0023). **d–f** qRT-PCR of the medaka pituitary of each genotype (**d**, *fshb*; **e**, *lhb*; **f**, *tshb*; *n* = 6 fish; ***p = 4.4e-4; **p = 0.0015). The data are mean ± SEM. ***p < 0.001, **p < 0.01, N.S. not significant, two-sided Dunnett's test (**b–f**). Source data are provided as a Source Data file.

recently, Ca²⁺ imaging of zebrafish FSH cells demonstrated the Ca²⁺ response to CCK peptide[41]. Moreover, in the Japanese eel, which is known to have diverged from other teleosts at the earliest stages of the teleost lineage[45], we demonstrated that *cck2r* mRNA is expressed in FSH cells (Supplementary Fig. 16). Therefore, the expression of CCK receptors in FSH cells appears to be widely conserved, at least in teleosts.

Furthermore, regarding the CCK ligand, some immunohisto-chemical studies have suggested the existence of CCK immunor-eactive neurons that project to the pituitary in the hypothalamus of other teleosts[31,32]. Additionally, the fact that hypophysiotropic CCK neurons co-expressed the paralogous *ccka* and *cckb* in medaka, as shown in the present study, strongly suggests that the enhancer that enabled the expression of CCK in the hypophysiotropic neurons was acquired before the divergence of *ccka* and *cckb* genes, which occur-red in the teleost-specific whole genome duplication[46]. This can be further interpreted that the ancestral vertebrates, before the emer-gence of teleosts, also possessed hypophysiotropic CCK neurons in their hypothalamus. These lines of evidence suggest that the mechanism proved in this study is widely conserved, at least in teleosts and perhaps even in other lineages. It is also worth noting that the CCK receptor is highly expressed in the chicken pituitary[47], although the

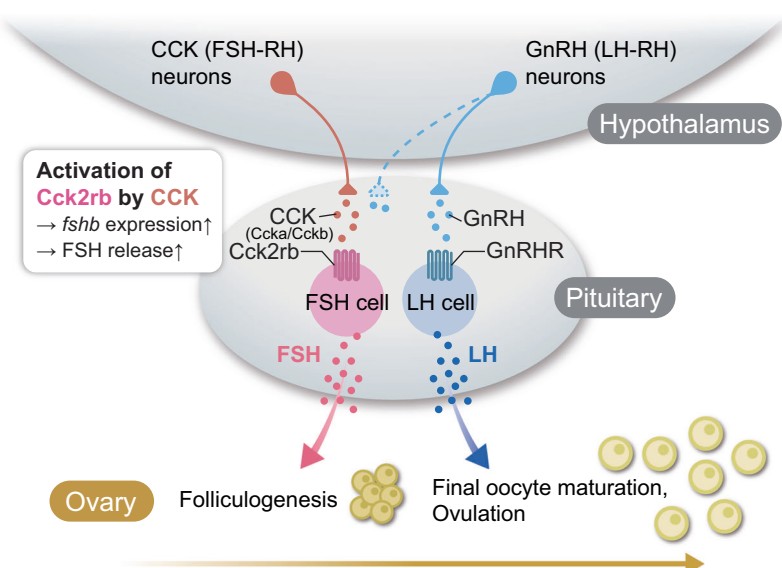

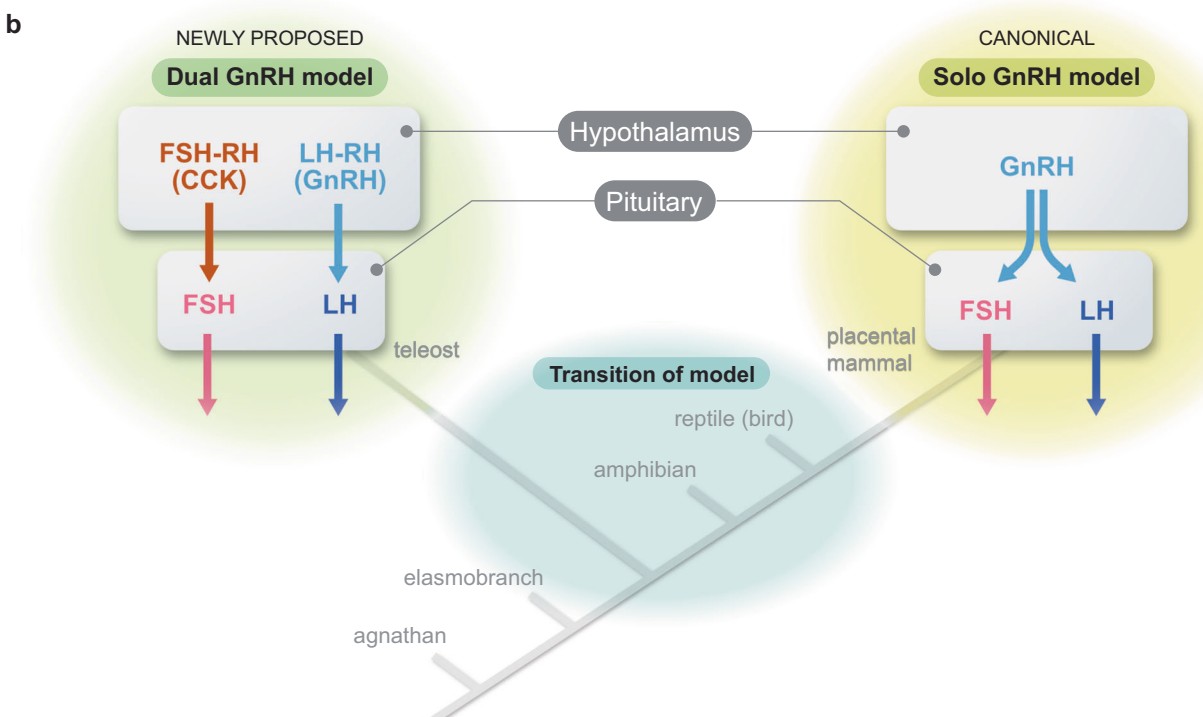

**Fig. 6 | Working hypothesis of the "dual GnRH model" in which two gonado-tropins are regulated by two gonadotropin-releasing hormones, FSH-RH and LH-RH. a** The schematic of the present study identifies CCK as the hypothalamic FSH-RH, which stimulates the expression and release of FSH to induce folliculo-genesis. After folliculogenesis is completed, GnRH (LH-RH)-induced LH induces final oocyte maturation followed by ovulation. **b** Illustration of the phylogenetic relationships indicating a transition of the models occurring during vertebrate evolution. We propose a "dual GnRH model" group in vertebrates, in addition to the canonical solo GnRH model group.

expressing cell types have not been identified. These facts strongly suggest that the conservation of the "dual GnRH model" is not restricted to a small subgroup of teleosts.

On the other hand, in placental mammals, there is currently no evidence to refute the established belief that they follow the solo GnRH model. In rodents, CCK does not induce FSH secretion in rats[48], and CCK-deficient mice are fertile[49]. Furthermore, the *Gnrh*-deficient mouse shows a reduction in serum FSH concentration[50], indicating

that LH-RH is an essential regulator for FSH as well as LH in this group. However, it should be inherently complex for the solo regulator to differentially regulate the release of distinct hormones. Perhaps these multiple roles of GnRH could be accomplished through the action of kisspeptin neurons. These neurons facilitate multi-modal GnRH secretion patterns, specifically the pulsatile and surge modes[51,52].

While it is generally understood that the regulation of GnRH neurons by kisspeptin is unique to mammals, as birds lack

kisspeptin-related genes[25,53] and knockout teleosts can reproduce[54,55], the mechanism behind the differential secretion of FSH and LH in all non-mammalian vertebrates remains unexplained. Thus, the "dual GnRH model" could potentially solve the remaining questions in species beyond just teleosts. Moreover, there exists an intriguing report indicating CCK-induced FSH release in human pituitary tumors[56], implying the possibility that regulation of FSH by CCK may have existed in our ancestors.

The discovery of CCK as FSH-RH could answer various remaining questions regarding the reproductive systems of teleosts. It is well-known that there are three paralogs of *gnrh* in vertebrates[57–60], and usually one of them that is expressed in hypophysiotropic neurons is responsible for gonadotropin release (e.g. *gnrh1* in medaka; *gnrh3* in zebrafish)[61,62]. However, in zebrafish, knocking out *gnrh3*, which is responsible for gonadotropin release, or even all of their *gnrh* genes, *gnrh2* and *gnrh3*, did not affect fertility[11,63]. The occurrence of folliculogenesis in them can be explained by FSH-RH playing a role in regulating FSH release in the absence of GnRH, which is similar to that of medaka. For their ability of final oocyte maturation and ovulation, other compensatory factors might be taken into consideration[64–67]. Nonetheless, the new system found in the present study may provide hints toward understanding such questions.

The *ccka/cckb* double homozygote KO reduced the expression level of *fshb* to <7% of *fshb* expressed compared to the double heterozygote KO of the same batch (Fig. 5), whereas *cck2rb* homozygote KO resulted in ~0.5% of the heterozygote KO (Fig. 1). This difference suggests the existence of a stimulator of Cck2rb other than CCK. A candidate for the compensatory factor is Gastrin, which has comparable biological activity to the CCK receptor (Fig. 3a). In PCR analysis, *gast* mRNA was detected in the gut rather than the hypothalamus (Supplementary Fig. 8). Therefore, this possible stimulator may come from the gut, which implies gut-derived CCK and Gastrin affect FSH release. However, it should be noted that this effect should be much weaker than that from hypophysiotropic CCK neurons, which directly innervate the FSH cells (Fig. 3e, f).

Similar to the co-expression of *ccka* and *cckb*, paralogous genes sometimes remain co-localized after gene duplication for a relatively long evolutionary period[68,69]. Further elucidation of the meaning of this co-expression of *ccka* and *cckb* could be an intriguing topic for future research.

Identification of the FSH-RH in the present study paves the way for new research avenues in neuroendocrinology. Given the evidence in mammals and teleosts, there should be a transition between the "dual GnRH model" and "solo GnRH model" in vertebrate evolution (Fig. 6b). It is crucial to understand which model the common ancestor of mammals and teleosts possessed. Further examination of vertebrate species should reveal the origin and evolutionary trajectories of the dual GnRH model and the solo GnRH model in vertebrates.

## Methods
### Animals
Medaka (*Oryzias latipes*) were maintained and used in accordance with the guiding principle for the Use and Care of Experimental Animals of the University of Tokyo. Male and female wild type and genetically modified medaka (himedaka strain used for FSH-cell specific RNAseq and d-rR strain used for all other experiments) were maintained under a 14-h light and 10-h dark photoperiod at a water temperature of 28 °C. The himedaka strain was acquired from a commercial source and the d-rR strain was acquired from National BioResource Project, Japan. The fish were fed at least twice a day with live brine shrimp and flake food except for Sundays. Medaka ranged from 3 to 6 months were used. In each subsequent experiment, siblings raised under the same conditions were used to control for genetic and environmental factors. Japanese eels (*Anguilla japonica*) of unknown age were purchased from commercial sources. Experiments were conducted in accordance

with the protocols approved by the animal care and use committee of the University of Tokyo (Permission number, P22-6, P22-7).

### FSH-cell Specific RNAseq to find the expressed receptors
To find highly expressed receptors in FSH cells, we performed FSH-cell specific RNAseq using GFP-labeled transgenic medaka. First, we established a transgenic medaka whose FSH cells are labeled by EGFP. After injecting a plasmid construct containing ~2 kb of the 5' flanking region of *fshb* of medaka and medaka heat shock protein minimal promoter (0.8 kb), the transgenic founder was selected in the F1 generation.

By using this transgenic medaka, FSH cells were collected based on their GFP fluorescence (Supplementary Fig. 1). First, pituitaries were dissected and subjected to dissociation with 0.2 mg/ml collagenase in artificial cerebrospinal fluid (ACSF) for 30 min. By gently pipetting with a narrowed pipette tip, the pituitary cells were dissociated. Under an inverted fluorescent microscope (AxioVert 100, Zeiss) with a manipulator, GFP-positive cells were collected with a glass pipette. Up to ten GFP-positive cells were pooled to each biological replicate.

After collecting four samples, sequence samples for Illumina sequencers were prepared using NEBNext Single Cell/Low Input RNA Library Prep Kit for Illumina (New England Biolabs, Ipswich, MA) according to the manufacturer's instruction. Sequencing of the resulting sequence samples was outsourced to a commercial sequencing service using Novaseq 6000 (Nippon Genetics, Tokyo, Japan). Sequence data were subjected to mapping and read count by STAR (2.5.4b) and R-SEM (1.3.3), respectively. The genes of receptors were selected and sorted in descending order.

### Single and double in situ hybridization
Adult female and male d-rR wild type medaka were deeply anesthetized with 0.02% tricaine methanesulfonate (MS-222). Since similar results were obtained in females and males, we represented females as representative data. Cryosections were prepared at 25 μm with a cryostat (Leica CM3050, Leica, Wetzlar, Germany; objective temperature: −24 °C, chamber temperature: −28 °C). The sliced sections were placed on the coated slide glass (CREST-coated, Matsunami, Kishiwada, Japan).

To confirm the phylogenetic relationship of receptors and the ligands of interest, we analyzed the deduced amino acid sequences of various species. After the acquisition of sequences by ORTHOSCOPE[70], phylogenetic trees were constructed using MEGA11[71] by the maximum likelihood tree method. The deduced amino acid sequences were aligned using GeneDoc 2.7.000 (National Resource for Biomedical Supercomputing, Pittsburgh, PA).

We analyzed the distribution of *cck2rb*, *ccka*, and *cckb* messenger RNA (mRNA) by in situ hybridization (ISH). Also, double in situ hybridization was performed to examine the co-localization of two genes. Single and double in situ hybridization was conducted. First, *cck2rb*-, *ccka*-, and *cckb*-specific digoxigenin (DIG)-labeled mRNA probes, and *cckb*- specific Fluorescein-labeled mRNA probes were prepared based on the cDNA region amplified by the primers listed in Supplementary Table 3 (Gene names in Ensembl: *ccka*, ENSORLG00000005949; *cckb*, ENSORLG00000005594; *cck2rb*, ENSORLG00000017966).

We also used a DIG-labeled and Fluorescein-labeled *fshb* probe according to the manufacturer's instruction. The signal of a single ISH was detected by an Anti-Digoxigenin AP antibody (1:2500; Sigma Aldrich, St. Louis, MO) and was visualized by NBT/BCIP. After the coverslip, we observed the sections through a BX53 biological microscope (Olympus, Tokyo, Japan). It was confirmed that signals were observed in slides hybridized with antisense probes and were not observed in sense probes (Supplementary Fig. 17). We followed the Medaka Histological Atlas for the nomenclature of the medaka brain nuclei.

In double in situ hybridization the fluorescein-labeled probe (*cck2rb* or *ccka*) was labeled by Anti-Fluorescein-HRP Conjugate (Perkin Elmer, Waltham, MA). The sections were subjected to TSA reaction using TSA plus biotin (Akoya Bioscience, Marlborough, MA) and the biotin signal was visualized by Streptavidin, Alexa Fluor 488 (Green; Thermo Fisher, Waltham, MA) after signal amplification with ABC elite kit (Vector Laboratories, Burlingame, CA). Then the first peroxidase activity label on the fluorescein probe was completely quenched with 3% $H_2O_2$ for 40 min before the application of an antibody to the DIG probe (1:500). The expression of *fshb*, *lhb*, or *cckb* was detected using DIG-labeled probes. The hybridized DIG-labeled probes were further detected by anti-digoxigenin-POD Fab fragments (Sigma Aldrich). The POD activity was then visualized by TSA plus Cy3 (Red; Akoya Bioscience).

The same protocol was applied to Japanese eel single/double in situ hybridization with minor modifications. The body weight of Japanese eels was 260-290 g. Probes were based on the database in the National Center for Biotechnology Information (NCBI). NCBI reference sequences are as follows: *eel_fshb*, XM_035417932.1; *eel_cck2r*, XM_035394241.1. The primers used to prepare probe templates are listed in Supplementary Table 3. We observed the signals through a confocal microscope, FV-1000 (Olympus) or Leica TCS SP8 (Leica Microsystems).

## Immunohistochemistry

Adult female and male d-rR wild type medaka were deeply anesthetized and their brains were fixed with 4% paraformaldehyde (PFA) in PBS and cryosectioned. Females are used as representative data in the main figure. Their body weight is 0.11–0.15 g.

To label the CCK-expressing cells, we used an anti-cholecystokinin (26-33) antibody raised in rabbit (C2581, 1:5000; Sigma Aldrich). After antigen retrieval with HistoVT (Nacalai Tesque, Kyoto, Japan) according to the manufacturer's protocol, a primary antibody was applied with 5% normal goat serum. After incubation with anti-rabbit IgG, a secondary antibody, signal amplification with an ABC Elite kit (Vector Laboratories, Burlingame, CA) was applied. Immunoreactivities were visualized with Streptavidin, Alexa Fluor 488 conjugate (1:500; Thermo Fisher). Some of the samples were labeled with 4′,6-diamidino-2-phenylindole (DAPI). We observed the signals through a confocal microscope FV-1000 (Olympus) or Leica TCS SP8 (Leica Microsystems).

## Dual labeling of biocytin and ccka mRNA

Adult female and male *fsh*:Inverse Pericam transgenic medaka (*fsh*:IP)[33] were used in this experiment to visualize the location of FSH cells. Females are shown in the figure as the representative data of both sexes. They were deeply anesthetized with 0.02% MS-222. The fish were decapitated, and the brains were excised. The fluoresced area (around FSH cells) of the pituitary was carefully injected with biocytin using the glass needle prepared from a glass capillary (G-1.5; Narishige, Tokyo, Japan) with a micropipette puller (P-97; Sutter Instruments, Novato, CA).

The brain was incubated with artificial cerebral spinal fluid (ACSF) containing: 134 mM NaCl, 2.9 mM KCl, 1.2 mM $MgCl_2$, 2.1 mM $CaCl_2$, 10 mM HEPES, and 15 mM glucose (pH 7.4, adjusted with NaOH) for 30 min and fixated with 4% PFA in PBS, then 30% sucrose in PBS and cryosectioned. The biocytin signal was amplified and visualized using an ABC kit and Streptavidin, Alexa Fluor 488 conjugate (1:500; Thermo Fisher). *ccka* mRNA signal was detected by ISH using a DIG-labeled probe and was visualized using an anti-DIG POD antibody (1:500) and TSA plus Cy3 (Akoya Bioscience).

## Generation of knockout medaka

*ccka*, *cckb*, and *cck2rb* knockout medaka were generated by CRISPR/Cas9-mediated genome editing. For *cck2rb* knockout, a CRISPR RNA (crRNA) was designed to target exon 1; for *ccka* knockout, two crRNAs were designed to target exon 2, which encodes the signal peptide of the Ccka precursor protein, and exon 3, respectively; for *cckb* knockout, two crRNAs were designed to target exon 3 and exon 4, which encodes the mature peptide of the CCK-8s, respectively. crRNA and trans-activating CRISPR RNA (tracrRNA) were synthesized by Fasmac (Kanagawa, Japan). Target sequences of the CRIPSR RNA including PAM are as follows. *cck2rbr*, AAGCGTGGACGGGTTCACGCAGG; *ccka*, TGACGCGTGTGATTGGTTAGTGG and ACCTGGGATGGATGGACTTTGGG; *cckb*, GGAGTGCTGGCCCTCATCTGAGG and GCAGCTGAAAGACCTTCCCGGGG. The crRNA, tracrRNA, and Cas9 protein (Nippon Gene Co. Ltd., Tokyo, Japan) were co-microinjected into medaka embryos at the one-cell or two-cell stage. Potential founder fish were screened by outcrossing with wild type fish and testing progeny for mutations by direct sequencing.

For the *cck2rb* knockout line, a founder was identified that produced progeny carrying an 8-bp deletion that caused a frameshift leading to complete loss of transmembrane domains. For the *ccka* knockout line, a founder was identified that produced progeny carrying a 743-bp deletion that caused a frameshift leading to premature truncation of the Ccka precursor protein. For the *cckb* knockout line, a founder was identified that produced progeny carrying a 210-bp deletion that caused a frameshift leading to premature truncation of the Cckb precursor protein. These progenies were intercrossed to establish knockout lines.

*ccka*/*cckb* double-knockout medaka were generated by crossing the progenies from the *ccka* knockout line and the *cckb* knockout line. Each line was maintained by breeding heterozygous or double-heterozygous individuals to obtain wild type, heterozygous, and knockout siblings for experimental use. The genotype of each fish was first determined by direct sequencing and thereafter by PCR and high-resolution melting analysis (for *cck2rb* knockouts) or sequencing, or agarose gel electrophoresis (for *ccka* and *cckb* knockouts) using the primers listed in Supplementary Table 3. The exon-intron structure and the deletion are illustrated in Supplementary Figs. 18–20.

Analyses were performed in the stage when all wild type medaka have matured and the siblings started spawning (~3 months after hatch). In the *ccka* and *cckb* double knockout, we additionally analyzed in ~4 months after hatch. The females of *cck2rb*, *ccka*, and *cckb* KO were crossed with fertile male drR wild type to examine the number of eggs from the female.

## Rescue experiment of Cck2rb in FSH cells

A rescue construct that induces FLAG-tagged Cck2rb expression specifically in FSH cells was prepared as a plasmid vector as follows. *Cck2rb* was isolated from the *cck2rb*/pcDNA3.1 vector used in the reporter assay experiment by using the primers described in Supplementary Table 3. FLAG-tag sequence was added to the C-terminus using the primer. The Cck2rb-FLAG coding sequence was ligated downstream of *fshb* promoter[33]. The construct also contained the larval stage-specific globin enhancer (globinb4[72]) fused with DsRed-Express for efficient screening of the transgenic founders[18].

To generate *cck2rb*⁻/⁻ medaka with the rescue construct, eggs spawned from *cck2rb*⁺/⁻ medaka were injected with the construct. After injection, the eggs were screened by fluorescence in the embryonic blood cells, which were derived from the globinb4 promoter:DsRed-Express. Once the medaka hatched and matured, female transgenic medaka were paired with male *cck2rb*⁻/⁻. The offspring were subjected to genotyping and individuals with *cck2rb*⁻/⁻ were selected. The presence or absence of the rescue transgene was determined from the fluorescence in the embryonic blood. Analyses were performed using medaka more than two months after hatching.

To validate the expression in FSH cells, the medaka were anesthetized in 0.02% MS-222 and their brains were excised, fixated, and cryosectioned. The FLAG-tag signal was detected by IHC using an anti-DDDDK antibody (1:1,000; raised in rabbit, PM020, MBL, Nagoya, Japan) and visualized using anti-rabbit IgG, Alexa 488 conjugate

(1:1,000; Invitrogen). The FSHb signals were also detected using an anti-FSHb antibody (a generous gift from Dr. Ogiwara[41]; specificity has been validated[26]) and visualized using anti-mouse IgG, Alexa Fluor 555 conjugate (1:1,000; Invitrogen). The nuclei of the cells were visualized by nuclear staining using methyl green (1:1,250). The ovaries and pituitaries were isolated for quantifying GSI and qRT-PCR respectively.

### E2 measurement
Peripheral tissues (the caudal halves of the bodies) were collected from *cck2rb*$^{+/+}$ and *cck2rb*$^{-/-}$ females at 2–4.5 h after the onset of the light period, frozen at −80 °C, and homogenized with Micro Smash (Tomy Seiko Co. Ltd., Tokyo, Japan). Tissue lipids were extracted with diethyl ether. Tissue levels of E2 were determined using the Estradiol ELISA Kit (Cayman Chemical Company, Ann Arbor, MI). Peripheral levels of E2 were expressed as picograms per mg tissue weight.

### Gonadal size and histology
After recording the body weight of each of the males or females (from the *cck2rb* knockout line or *ccka*/*cckb* double-knockout line), the gonad was removed, weighed, and photographed under a stereo microscope M205FA (Leica Microsystems) equipped with a digital camera DFC7000T (Leica Microsystems). The gonadosomatic index (GSI) was calculated using the following formula. GSI = [gonad weight / total tissue weight] × 100. After being photographed, the gonads were fixed in 4% paraformaldehyde (PFA) and embedded in paraffin. Five-μm thick sections were cut and stained with hematoxylin and eosin. Images were acquired using the VS120 slide scanner (Olympus).

### Quantitative RT-PCR of pituitary
Generated knockouts and their siblings were anesthetized, and the pituitary was collected for real-time PCR analysis. Total RNA was extracted from each pituitary using a Fast Gene RNA Basic kit (Nippon Genetics) according to the manufacturer's protocol. Total RNA was reverse-transcribed with the PrimeScript RT kit (Takara, Kusatsu, Japan) according to the manufacturer's instructions.

For real-time PCR, the cDNA was amplified using a KAPA SYBR fast qPCR kit (Nippon Genetics) with LightCycler 480 II system (Roche, Mannheim, Germany). The temperature profile of the reaction was 95 °C for 5 min, 45 cycles of denaturation at 95 °C for 10 s, annealing at 60 °C for 10 s, and extension at 72 °C for 10 s. The PCR product was verified by melting curve analysis. A housekeeping gene, β-actin (*actb*) was used for normalization. Primers used in this experiment are shown in Supplementary Table 3.

### Reverse transcription PCR
Adult female d-rR wild type medaka were deeply anesthetized, and their brain and intestine were excised. RNA was extracted using the Fast Gene RNA Basic kit according to the manufacturer's protocol. The RNA was purified with DNaseI and reverse-transcribed with the PrimeScript RT kit according to the manufacturer's protocol. The cDNA was amplified using KAPA HiFi Hotstart ReadyMix PCR kit (Nippon Genetics) with T-100 Thermal Cycler (Bio-rad, Hercules, CA). The temperature profile of the reaction was 95 °C for 2 min, 35 cycles of denaturation at 95 °C for 10 s, annealing at 58 °C for 30 s, and extension at 72 °C for 10 s. The PCR products were verified by gel electrophoresis on 2% agarose gel. The primer pairs used in PCR are listed in Supplementary Table 3.

### Reporter activation assay
The medaka CCK-8s (DY(SO$_3$H)LGWMDF-NH$_2$) and Gastrin-8s (DY(SO$_3$H)RGWLDF-NH$_2$) peptides were synthesized by Scrum (Tokyo, Japan). The cDNA fragment encoding the full-length *cck2rb* was PCR-amplified and subcloned into the expression vector pcDNA3.1/V5-His-TOPO (Thermo Fisher Scientific). The primers used here are listed in Supplementary Table 3.

The resulting Cck2rb expression construct was transiently transfected into HeLa cells together with a luciferase reporter vector containing cis-acting elements responsive to cAMP (pGL4.29; Promega), Ca$^{2+}$-dependent nuclear factor of activated T-cells (NFAT) (pGL4.30; Promega), or the MAPK signaling pathway (pGL4.33; Promega) and the internal control vector pGL4.74 (Promega) using Lipofectamine LTX (Thermo Fisher Scientific). Forty-two hours after transfection, cells were stimulated with CCK-8s or Gastrin-8s polypeptide at doses of 0, 10$^{-11}$, 10$^{-10}$, 10$^{-9}$, 10$^{-8}$, 10$^{-7}$, or 10$^{-6}$ M for 6 h.

After cell lysis, luciferase activity was measured using the Dual-Glo Luciferase Assay System (Promega). Each assay was performed in duplicate and repeated three times independently. HeLa cells used in this study were authenticated by short tandem repeat profiling (National Institute of Biomedical Innovation, Osaka, Japan) and confirmed to be mycoplasma-free (Biotherapy Institute of Japan, Tokyo, Japan).

### Ca$^{2+}$ imaging
Adult *fsh*:IP and *lh*:IP were anesthetized in MS-222 and were decapitated. The pituitary was excised out to the recording chamber. Unless otherwise mentioned, female fish were used in this experiment. Their body weight was 0.152-0.224 g and 0.215 g for males. The medaka CCK-8s peptide was diluted to 0.1, 1, 10, 100, or 1000 nM with 0.01% DMSO in ACSF. We also diluted mdGnRH (synthesized by Genescript, Piscataway, NJ) to 100 nM with ACSF and used it for the comparison of the effect with CCK-8s peptides.

The pituitary was perfused using a peristaltic pump (Rainin Dynamax RP-1, Rainin, Columbus, OH), and images were taken every 5 s with a scientific complementary metal oxide semiconductor camera (scMOS) (Andor Zyla 4.2 PLUS, Oxford Instruments, Belfast, UK) configured to the fluorescent lamp (X-Cite 110 LED Illumination system, Excelitas Technologies, Waltham, MA). Images were captured by Micro-Manager 1.4. The pituitary was washed with ACSF for 10 min after CCK-8s was applied.

### Examining the effect of CCK on gonadotropin mRNA in vitro
Adult female d-rR wild type and generated *cck2rb* KO medaka were deeply anesthetized and their pituitaries were incubated. Their body weight was 0.17−0.36 g. We used 0, 10, or 100 nM medaka CCK-8s in 200 μL of culture medium Leibovitz L-15 medium (Thermo Fisher) supplemented with 1% 100x penicillin-streptomycin, 5% heat-inactivated fetal bovine serum (FBS), and 10 mM D-glucose.

The pituitaries were incubated at 27 °C for 48 h. After 48 h, the medium was removed, and qRT-PCR was conducted. For qRT-PCR, the cDNA was amplified using a KAPA SYBR fast qPCR kit (Nippon Genetics) with LightCycler 480 II system (Roche, Mannheim, Germany). The temperature profile of the reaction was 95 °C for 5 min, 45 cycles of denaturation at 95 °C for 10 s, annealing at 60 °C for 10 s, and extension at 72 °C for 10 s. The PCR product was verified by melting curve analysis.

Similar experiments using females (body weight: 0.17−0.25 g) and males (body weight: 0.12−0.27 g) were additionally performed with 0 or 100 nM medaka CCK-8s. The body weight of medaka used here was females. We also performed the same experiment using the pituitaries of hetero and homo *cck2rb* knockout females (body weight: 0.16−0.22 g for hetero, 0.15−0.36 g for homo). A housekeeping gene, β-actin (*actb*) was used for normalization. Primers used in this experiment are shown in Supplementary Table 3.

### Data analysis
All values are shown as mean ± standard error of the mean (SEM). Statistical tests were performed by two-sided. Groups of two were processed with Student's *t*-test, and groups of more than three were processed with Dunnett's test.

Microphotographs and images were processed by ImageJ software (National Institutes of Health, Bethesda, MD). Statistical analyses

were performed with Kyplot 6.0 (Kyence, Tokyo, Japan). Graphs were drawn with Kyplot 6.0 or R (R Foundation).

## Reporting summary

Further information on research design is available in the Nature Portfolio Reporting Summary linked to this article.

## Data availability

All raw sequencing data of medaka FSH cells generated in this study have been deposited in the NCBI BioProject database under accession number PRJDB16930. Other data generated in this study are provided in the main text or the Supplemental materials. Source data are provided in this paper. Source data are provided with this paper.

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

## Acknowledgements

We thank Dr. Shin-ichi Higashijima (National Institute for Basic Biology, Japan) for providing a plasmid containing the medaka heat shock promoter sequence. We are also indebted to Dr. Katsueki Ogiwara (Hokkaido University, Japan) for providing the antibody against medaka FSHβ. We also thank the staff at the Laboratory for Phyloinformatics, RIKEN BDR for helpful advice on low-input RNA-seq technique. We are grateful to Dr. Daichi Kayo (Tohoku University, Japan) for the helpful discussion and comments. We also thank Dr. Soma Tomihara (Nagahama Institute of Bio-Science and Technology, Japan) for help in the construction of plasmid DNA. This work was funded by the Japan Society for the Promotion of Science for S.Ka and K.O. (23H02306) and for S.Ka (18K19323, 18H04881), and by Mitsubishi Foundation, Mishima Kaiun Memorial Foundation, and Sumitomo Foundation for S.Ka.

## Author contributions

S.K.U., Y.N., K.O., and S.Ka conceived the project. S.K.U., Y.N., and S.Ka performed histological analyses, while S.K.U. performed Ca²⁺ imaging and in vitro analysis of the pituitary. Y.N. analyzed the knockouts and reporter assay. S.Ka generated GFP transgenic medaka. K.M. performed

FSH cell-specific RNA-seq. T.K. and S.Ku helped with the methodology of FSH cell-specific RNA-seq. K.O. and S.Ka assisted with data interpretation. S.K.U., Y.N., and S.Ka wrote the original draft. All authors contributed to the editing of the manuscript.

## Competing interests

K.O. and S.Ka (The University of Tokyo) have filed a patent related to this study (Japanese Patent Application No. 2023-25897). The remaining authors declare no competing interests.
