## [Peer Review File · Nature Communications]

Identification of the FSH-RH as the other gonadotropin-releasing hormoneREVIEWER COMMENTS

Reviewer #1 (Remarks to the Author):

Using medaka as the model, this study addressed the issue of what neuroendocrine factor(s) is responsible for regulating pituitary FSH biosynthesis in fish. Although GnRH has been well documented to regulate both FSH and LH in mammals, mainly by varying frequencies, how the two gonadotropins are differentially regulated in fish remains largely unknown. This question becomes even more perplexing after the reports in both zebrafish and medaka that the knockout mutant lines of GnRH showed normal gonadal development. Using elegant genetic approach, together with in vivo and in vitro experiments, this study provides convincing evidence that the pituitary FSH cells are subject to regulation by another neuropeptide CCK. The authors first demonstrated that the FSH cells expressed abundant CCK receptor *cckbr1* by FSH cell-specific RNA-seq analysis. They then generated a *cckbr1* mutant line, which showed significantly smaller gonads in both females and males. Both *fshb* and *lhb* showed a significant decrease in their expression, but the reduction of *lhb* expression seemed to be due to the reduced steroid feedback from the mutant gonads. They then tested the effects of potential ligands of *Cckbr1*, including CCK8 and gastrin, using a receptor reporter assay. The role of CCK8 in controlling FSH cells was further confirmed in knockout lines of CCK genes (*ccka* and *cckb*). Further evidence for CCK8 regulation of FSH cells was obtained from an in vitro experiment showing that CCK8 stimulated signaling pathways in labelled FSH cells and *fshb* expression in the incubated pituitary glands. This was supported by the lack of response of the pituitary glands from *cckbr1* mutant fish, and the evidence that the double knockout of *ccka* and *cckb* phenocopied the mutant of *cckbr1*. In general, this is a well-designed study involving elegant genetics approach. The data obtained are comprehensive and support the conclusion. The discovery of this study represents a major advancement in our understanding of fish reproductive endocrinology. This reviewer has the following points for authors to consider during revision.

1. The authors proposed that the fish have so-called "dual GnRH model" instead of "solo GnRH model" as in mammals. The data provided in the manuscript do support such argument. However, for this new model to be accepted, we would need more evidence. One critical piece of evidence that does not seem to support the CCK-FSH and GnRH-LH model is that the FSH cells also expressed high level of GnRH receptor (*gnrh-r2*), suggesting potential regulation of FSH cells by GnRH as well in addition to CCK8.
2. To address the issue above, the authors should carry out parallel studies on GnRH as they did for CCK8.
3. In all assays on bioactivities, gastrin showed higher potency in stimulating *Cckbr1* than CCK8. This was not discussed in the manuscript.
4. Both CCK and gastrin were expressed in the intestine. How do we know that it is the brain but not intestinal CCK that plays a critical role in regulating FSH cells?
5. Was there colocalization of *lhb* and *cckbr1* expression?
6. In the assay on Ca²⁺ in FSH cells, 1 μM CCK8 was used. For neuropeptides, 1 μM (1000 nM) is most likely beyond the physiological concentration range. Were lower concentrations tested? If yes, what were the responses?
7. Fig. 3 CCK8 also seemed to increase *lhb* expression in a dose-dependent manner despite lack of statistical significance. An increase in sample size may help.
8. Fig. 4 Double knockout of *ccka* and *cckb* caused significant decrease in both *fshb* and *lhb*. Was the response of *lhb* a secondary response again?
9. L185: What was the evidence for the compensatory mechanism involving LH action on FSH receptor in medaka? This has been reported in zebrafish. The authors should refer to the studies in zebrafish demonstrating cross-reactivity of LH with FSHR and deletion of *fshb* gene resulted in an increased expression of *lhb*.
10. L172: change "Kos" to "KOs"
11. L178: change "smaller" to "lower"
12. L251 and 257: italicize the species names

13. L265: italicize fshb
14. Supplementary Fig. 3: delete one "of" in the 3rd line
15. Supplementary Fig. 9: what about co-expression of lhb and cckbr?

Reviewer #2 (Remarks to the Author):

In this manuscript the authors propose the existence of a "dual GnRH model" where cholecystokinin is the FSH-Releasing Hormone (not GnRH as in the generally accepted model), and GnRH is the LH-RH. The findings are intriguing, but as they stand, several points need to be further addressed to support their hypothesis that CCK is FSH-RH in the intact animals. Namely a more careful description of the CRISPR/Cas9 mutant animals specifically the fertility and fecundity assayed through crosses, a more careful analysis of the role of GnRH in fishes, and the lack of specificity of the antibodies used. Authors need to more fully address the observation presented at the end of the results on the fertility of the mutants. Authors need to temper their language throughout the manuscript (challenge versus investigate, prove versus support etcetera).

GENERAL:

1. PLEASE CHECK ENGLISH THROUGHOUT THE MANUSCRIPT INCLUDING THE METHODS.
2. WHY DO AUTHORS USE +/- AS CONTROLS: WILD TYPE ANIMALS SHOULD BE INCLUDED
3. AUTHORS NEED TO CLEARLY STATE THROUGHOUT THE TEXT THE SOURCE OF ALL PEPTIDE AND ANTIBODIES. ARE THE PEPTIDES FROM FISH?
THE CCK ANTIBODY THEY USED IS MADE AGAINST HUMAN PROTEIN THUS THE AUTHORS SNEED TO CALL THE LABELLING "CCK-LIKE"
4. AUTHORS NEED TO FULLY EXPLAIN GNRH IN FISHES (3 FORMS IN GENERAL) AND THE KO-PHENOTYPE IN ZEBRAFISH.
5. AUTHORS NEED TO EXPLAIN WHY THEY DID NOT FOLLOW-UP OIN THE FINDING OF GNRH-R2 IN THE RNASEQ ANALYSIS.
6. ELIMINATE THE DATA ON THE JAPANESE EEL, IT IS NOT FULLY EXPLORED AND AGAIN THE ANTIBODIES ARE NOT AGAINST THE PEPTIDE OF THIS ANIMAL.

TEXT:

Therefore, to challenge the
29 canonical "solo GnRH model," we aimed to identify the other gonadotropin regulator,
CHANGE WORDING: replace challenge with investigate

In the present study, to challenge the current "solo GnRH model," we aimed to identify
65 the FSH-RH.

CHANGE WORDING: replace challenge with investigate

However, this once-established consensus has been challenged in vertebrates other than
55 mammals. Intriguingly, it has been reported that GnRH knockout (KO) does not affect FSH
56 function in model teleosts such as medaka and zebrafish 10,11, which implies that GnRH may not
57 be the primary regulator of FSH release, at least in teleosts.

AUTHORS NEED TO MORE CLEARLY EXPLAIN THESE PAPERS, AS THE SENTENCE IS WRITTEN IT
SOUNDS LIKE THE KO ONLY AFFECTS FSH FUNCTION, THIS IS NOT TRUE: THE ANIMALS ARE FULLY
FERTILE. ALSO EXPLAIN THE DIFFERENT FORMS OF GNRH (MEDAKA HAVE THREE, ZEBRAFISH HAVE
2)

75 Among the metabolic receptors expressed, we found that cholecystokinin B receptor 1 (cckbr1)76
had the highest expression (Supplementary Table 1).

WHY DO AUTHORS DISCOUNT GNRH-R2, WHICH IS THE SECOND HIGHEST EXPRESSION?

Thus, we 105 proved that *Cckbr1*, which is expressed in FSH cells, is essential for the normal function of FSH

AGAIN "PROVED" IS A VERY STRONG WORD,

differ among all genotypes (Fig. 4d, e). The fact that only the double KO was associated with a 181 severe phenotype suggests that *ccka* and *cckb* have redundancy in their function of FSH
CROSSES ARE NEEDED HERE TO CONFIRM FERTILITY DEFECTS

183 hypophysiotropic neurons (Fig. 2). Interestingly, some of the double knockouts started spawning 184 one month after the wild type started to spawn even with completely reduced *fsbh* expression and 185 significantly smaller ovary ($n = 4$), which can be explained by some compensatory mechanism 186 that involves LH action on FSH receptor (Supplementary Fig. 8). These results indicate that 187 cholecystokinin is the primary factor responsible for the expression and release of FSH, which 188 should be referred to as the FSH-RH.

THIS IS A WORRYING OBSERVATION AND IS WHY ALL GENOTYPES NEED TO BE CROSSED AND THE NUMBER OF EMBRYOS SCORED, AND THESE DATA INCLUDED IN THE MANUSCRIPT

METHODS

To label the CCK-expressing cells, we used an anti-cholecystokinin (26-33) antibody raised in 316 rabbit (C2581, Sigma-Aldrich, St. Louis, MO; 1:5,000).

THIS IS A HUMAN SPECIFIC ANTIBODY: AUTHORS NEED TO CHANGE THE WORDING THROUGHOUT THE TEXT OF THE MANUSCRIPT TO "CCK-LIKE"

Double in situ hybridization was visualized by TSA plus biotin

301 followed by ABC Elite kit (Vector Laboratories, Burlingame, CA) and Streptavidin, Alexa 302 Fluor 488 (Green; Thermo Fisher, Waltham, MA), and TSA plus Cy3 (Red; Akoya Bioscience, 303 Marlborough, MA). Note that the first peroxidase label on the fluorescein probe was completely 304 quenched with 3% H₂O₂ for 40 mins before the second antibody for the DIG probe was labeled.
THIS SECTION NEEDS TO BE RE-WRITTEN. AS IT APPEARS HERE THE TWO DIFFERENT FLUORESCENT FLUOROCHROMES IN THE ISH ARE UNCLEAR: PLEASE STATE CLEARLY WHICH PROBES WERE LABELED WITH DIG AND WHICH WERE LABELLED WITH FLUORESCENT LABELLED PROBES. DID THE AUTHORS USE ANTI-DIG AND ANTI-FLUORESCIN? IT LOOKS LIKE THE AUTHORS COPIED THE SECTION FROM IMMUNOCYTOCHEMISTRY FOR ANTIBODIES.

340 Generation of knockout medaka

341 *ccka*, *cckb*, and *cckbr1* knockout

- SUPP FIGURE 11 DIAGRAM: EXPLAIN WHAT THE GREEN AND BLUE COLORS MEAN
- PLEASE INCLUDE ALIGNMENT OF SEQUENCES FROM THE WILD TYPE AND MUTANT GENES (REGION THAT IS DELETED), ESPECIALLY IMPORTANT FOR THE *CCKBR1* SEQUENCE.
- PLEASE GIVE A MORE THOROUGH DESCRIPTION OF THE MUTANTS: DATA NEED TO BE PRESENT FOR CROSSES OF THE FISH TO SHOW THAT THEY ARE TRULY AFFECTED BY THE MUTATION. WHAT IF THE OVARIES ARE REDUCED IN SIZE BUT FISH STILL CAN LAY A FEW EGGS.
- ARE THEY FERTILE AS HETEROZYGOTES?
- WHAT HAPPENS WHEN YOU CROSS THESE KO FISH?

THESE DATA NEED TO BE SHOWN FOR EACH KO (SEE OTHER EXAMPLES IN THE LITERATURE, LIKE ZOHAR LAB)

ALSO IT APPEARS THE AUTHORS INJECTED ON A SINGLE GUIDE RNA FOR EACH GENE. IF THIS IS TRUE THEY NEED TO EXPLAIN WHY THEY GENERATED MUTANTS WITH SUCH LARGE DELETIONS: *CCKA* 743-bp deletion, *CCKB* 210-bp deletion

FIGURES:

Authors make statements about fibers when the figures are of too low magnification to draw

conclusions (see Supp Fig 5)

Supp Fig 7 has a clear sex bias in the male versus female data: why is there no assay of ovary weight?

SUpP fig 8: confusing: if ccka/b double knock shows no phenotype (See Panels in a), how can authors conclude it controls FSH (which controls EARLY gonad development)?

Supp figure 9 is expression in the Japanese eel (?)

We found the Referees' comments very constructive and helpful for us in revising the manuscript. We have studied your comments very carefully and have made the necessary emendations.

The point-by-point responses to the reviewers' comments are listed below. The paragraph written after "Response" denotes our response. In the main text, the modified texts are highlighted as yellow.

Reviewer #1

1. The authors proposed that the fish have so-called "dual GnRH model" instead of "solo GnRH model" as in mammals. The data provided in the manuscript do support such argument. However, for this new model to be accepted, we would need more evidence. One critical piece of evidence that does not seem to support the CCK-FSH and GnRH-LH model is that the FSH cells also expressed high level of GnRH receptor (*gnrh-r2*), suggesting potential regulation of FSH cells by GnRH as well in addition to CCK8.

Response:

We thank the reviewer for the helpful comment. We agree with the reviewer's point that GnRH also affects FSH secretion, although the effect and essentiality of GnRH are much weaker than that of CCK. Accordingly, we have added a description of experiments in the main text with additional experimental data, which is described in detail in the response to the next comment (comment "2.").

2. To address the issue above, the authors should carry out parallel studies on GnRH as they did for CCK8.

Response: We performed an additional experiment of Ca^{2+} imaging using the pituitary of *fsh:IP* and perfusing it with 100 nM GnRH and 100 nM CCK8. In this experiment, as suggested by the reviewer and in our previous study (Karigo et al., 2014), GnRH affected $[\text{Ca}^{2+}]_i$ of FSH cells. Simultaneously, the effect of GnRH was weaker than CCK8 of the same concentration (Supplemental Fig.8).

Supplementary Fig. 8. $[\text{Ca}^{2+}]_i$ of FSH cells when perfused with 100 nM mdGnRH or 100 nM CCK8 peptide.

a Ca^{2+} imaging of FSH cells applied with 100 nM mdGnRH or 100 nM CCK8

b The response of FSH cells to CCK8 peptide is significantly greater than that to mdGnRH at 100 nM ($n=5$). $*P<0.05$.

Accordingly, we revised the manuscript as follows:

Additionally, we examined the Ca^{2+} response when perfused with GnRH1 (mdGnRH)³², which is considered to be the intrinsic GnRH subtype controlling gonadotropin secretion^{26,33-35}. Although mdGnRH (100 nM) induced a Ca^{2+} response in FSH cells as suggested in a previous study³⁰, this response was significantly lower than that induced by CCK8 at the same concentration ($n = 5$, Supplementary Fig. 8a, b). Therefore, we suggest here that CCK can be the primary hypophysiotropic factor to regulate FSH release.

3. In all assays on bioactivities, gastrin showed higher potency in stimulating Cckbr1 than CCK8. This was not discussed in the manuscript.

Response: We thank the reviewer's valuable comment. As the reviewer mentioned, our luciferase assay indicated that Gastrin showed more potent activation on Cckbr1-expressing HeLa cells. On the other hand, the possibility of Gastrin regulation on FSH secretion was proven to be very low because there was no detectable expression of *gast* (*gastrin*) in the hypothalamus. This was consistent with the fact that *ccka* and *cckb* double knockout medaka, which retain the *gast* gene, showed drastically reduced *fshb* expression. We added the explanation in the main text as follows:

The *ccka/cckb* double homozygote KO reduced the expression level of *fshb* to ~6% of *fshb* expressed compared to the double heterozygote KO of the same batch (Fig. 5), whereas *cckbr1* homozygote KO resulted in ~0.5% of the heterozygote KO (Fig. 1). This difference suggests the existence of a stimulator of Cckbr1 other than CCK. A candidate for the compensatory factor is Gastrin, which has comparable biological activity to the CCK receptor (Fig. 3a). In PCR analysis, *gast* mRNA was detected in the gut rather than the hypothalamus (Supplementary Figure 5). Therefore, this possible stimulator may come from the gut, which implies gut-derived CCK and gastrin affect FSH release. However, it should be noted that this effect should be much weaker than that from hypophysiotropic CCK neurons, which directly innervate to the FSH cells (Fig. 3e,f).

4. Both CCK and gastrin were expressed in the intestine. How do we know that it is the brain but not intestinal CCK that plays a critical role in regulating FSH cells?

Response: Thank you for the insightful comment. Because of the following reasons, we consider hypothalamic CCK to be the primary factor that regulates FSH release. First, in *ccka/cckb* double knockout, *fshb* expression in the pituitary was only ~6% of the double hetero siblings, which indicates that CCK but not gastrin is the primary factor in regulating FSH (Fig.5d, Supplemental Fig. 12). If gut-derived CCK is the main regulator, such drastic change should only be observed when both CCK and gastrin are knocked out, as both are released from the gut.

Secondly, the dense fiber projection of CCK neurons to the FSH cells indicates that this innervation is the primary source of ligands for Cckbr1. Moreover, intestinal CCK/gastrin is only released temporarily when food comes into the gut with a half-life of 1.3-4.4 minutes (in dogs; Hoffmann, 1993). Therefore, intestinal CCK/gastrin should not have a more significant effect compared to hypothalamic CCK, which directly innervates the FSH cells.

Based on these reasons, although there is a possibility that intestinal CCK/gastrin

may affect FSH release, we conclude that hypothalamic CCK neurons are the main players in FSH regulation. We have added this explanation in the main text, as was answered in the previous comment.

5. Was there colocalization of *lhb* and *cckbr1* expression?

Response: Thank you for the important question. We agree that the additional data examining the co-expression of *lhb* and *cckbr1* in the pituitary of the medaka strengthens our hypothesis. Therefore, we performed another double *in situ* hybridization and found that there was no colocalization of *lhb* and *cckbr1* (Fig.1b).

This result is consistent with the results of Ca^{2+} imaging, in which LH cells did not respond to CCK application. The explanation has been added in the main text as follows: Therefore, we conducted double *in situ* hybridization of the FSH subunit beta gene (*fshb*) or LH subunit beta gene (*lhb*) and *cckbr1* to examine their co-expression in the pituitary through the histological method. From these experiments, we demonstrated that *cckbr1* is expressed exclusively in FSH cells (Fig. 1a).

Also, there was no co-localization of *lhb* and *cckbr1* in the pituitary of Japanese eel, which suggests the conservation of this property in teleosts (added in the revised manuscript as Supplementary Fig. 13c).

6. In the assay on Ca²⁺ in FSH cells, 1 μ M CCK8 was used. For neuropeptides, 1 μ M (1000 nM) is most likely beyond the physiological concentration range. Were lower concentrations tested? If yes, what were the responses?

Response: As the reviewer pointed out, 1 μ M CCK8 is a very high concentration. However, we used this concentration to show that even at this concentration, CCK does not affect LH cells. For FSH cells, we used 0.1, 1, 10, and 100 nM and observed the dose-dependent curve of the fluorescence response, which has been shown in the original manuscript (Fig. 4b of the revised version).

7. Fig. 3 CCK8 also seemed to increase *lhb* expression in a dose-dependent manner despite lack of statistical significance. An increase in sample size may help.

Response: As the reviewer pointed out, it appears that there is a possible effect. To address this, we performed an additional experiment containing 0 nM and 100 nM (Supplementary Fig. 9), however, no statistical difference was observed. Therefore, we concluded that we do not see the effect of CCK8 on *lhb* expression. This is consistent with the lack of expression of *Cckbr1* in LH cells proved by another additional experiment (Supplementary Fig. 9).

Supplementary Fig. 9. CCK8 exclusively increases the *fshb* expression.

Since there might be an increasing trend in *lhb* expression after incubation with CCK8, similar experiments were reexamined. **a-c** qRT-PCR of the pituitaries after incubating in CCK8 for 48 hours (a, *fshb*; b, *lhb*; c, *tshb*; $n = 5$). Only *fshb* expression was significantly different between the pituitary incubated with or without 100 nM CCK8. The data are mean \pm SEM. ** $P < 0.01$, N.S., not significant.

This explanation is included in the revised main text as follows:

Also, a similar experiment incubating the pituitary in 0 nM and 100 nM CCK8 ($n = 5$) did not show a significant difference in *lhb* expression (Supplementary Fig. 9). Considering the results of Ca²⁺ imaging and double *in situ* hybridization, it is highly probable that LH cells do not possess CCK receptors. It is therefore reasonable that we did not observe an increase of *lhb* expression.

8. Fig. 4 Double knockout of *ccka* and *cckb* caused significant decrease in both *fshb* and *lhb*. Was the response of *lhb* a secondary response again?

Response: As suggested by the reviewer, we consider that this effect is similar to the secondary effect observed in the *cckbr1* knockout. The decrease in *fshb* leads to a significant reduction in estrogen levels due to the immature ovary, resulting in a secondary effect of reduced *lhb* expression. Supporting this idea, the pituitaries of spawned DKO females showed *lhb* expression comparable to that of wildtype females, while *fshb* expression remained low (Supplementary Fig. 12). The explanation has been added in the revised manuscript as follows:

This can be explained by the compensatory mechanisms involving LH as follows. First, a small amount of FSH is released by another ligand of *Cckbr1*, perhaps Gastrin, which slowly stimulates folliculogenesis. This causes an increase in serum estrogen, which should induce LH secretion²⁴. Here, in medaka, because LH can also activate FSH receptor³⁸, LH can regulate the ovulatory cycle in an FSH-independent manner once LH secretion is activated. Despite the occurrence of this delayed spawning, the overall results consistently indicate a severe deficiency of the FSH system in *ccka/cckb* double KO. Therefore, cholecystokinin is the primary factor responsible for the expression and release of FSH, which should be referred to as the FSH-RH.

9. L185: What was the evidence for the compensatory mechanism involving LH action on FSH receptor in medaka? This has been reported in zebrafish. The authors should refer to the studies in zebrafish demonstrating cross-reactivity of LH with FSHR and deletion of *fshb* gene resulted in an increased expression of *lhb*.

Response: We thank the reviewer for an insightful comment. It has also been reported in medaka that LH cross-react with FSHR. We added a more detailed explanation with this citation in the revised manuscript. We also cited zebrafish studies to extend the generality of this phenomenon and added the explanation as follows:

It is well-known that there are three paralogs of *gnrh* in vertebrates⁵⁴⁻⁵⁷, and usually one of them that is expressed in hypophysiotropic neurons is responsible for gonadotropin release (e.g. *gnrh1* in medaka; *gnrh3* in zebrafish)^{58,59}. However, in zebrafish, knocking out *gnrh3*, which is responsible for gonadotropin release, or even all of their *gnrh* genes, *gnrh2* and *gnrh3*, did not affect fertility^{11,60}. The occurrence of folliculogenesis in them can be explained by FSH-RH playing a role in regulating FSH release in the absence of GnRH, which is similar to that of medaka. For their ability of final oocyte maturation and ovulation, other compensatory factors might be taken into consideration⁶¹⁻⁶⁴. Nonetheless, the new system found in the present study may provide hints toward understanding such

questions.

10. L172: change “Kos” to “KOs”

Response: The mentioned line was changed accordingly.

11. L178: change “smaller” to “lower”

Response: The mentioned line was changed accordingly.

12. L251 and 257: italicize the species names

Response: The mentioned line was changed accordingly.

13. L265: italicize *fshb*

Response: Thank you for the suggestion. The mentioned line was changed accordingly.

14. Supplementary Fig. 3: delete one “of” in the 3rd line

Response: We thank the reviewer for the helpful comment. Supplementary Fig. 3 was renamed to Supplementary Fig. 4, and was revised accordingly.

15. Supplementary Fig. 9: what about co-expression of *lhb* and *cckbr*?

Response: Thank you for the important question. Supplementary Fig. 9 was renamed to Supplementary Fig. 13, and DISH of *lhb* and *cckbr1* was added accordingly.

Reviewer #2

In this manuscript the authors propose the existence of a “dual GnRH model” where cholecystinin is the FSH-Releasing Hormone (not GnRH as in the generally accepted model), and GnRH is the LH-RH. The findings are intriguing, but as they stand, several points need to be further addressed to support their hypothesis that CCK is FSH-RH in the intact animals. Namely a more careful description of the CRISPR/Cas9 mutant animals specifically the fertility and fecundity assayed through crosses, a more careful analysis of the role of GnRH in fishes, and the lack of specificity of the antibodies used. Authors need to more fully address the observation presented at the end of the results on the fertility of the mutants. Authors need to temper their language throughout the manuscript (challenge versus investigate, prove versus support etcetera.

GENERAL:

1.PLEASE CHECK ENGLISH THROUGHOUT THE MANUSCRIPT INCLUDING THE METHODS.

Response: We thank you for the suggestion. In accordance with the reviewer, we have thoroughly reviewed the manuscript and revised it as necessary.

2. WHY DO AUTHORS USE +/- AS CONTROLS: WILD TYPE ANIMALS SHOULD BE INCLUDED

Response: Thank you for the important comment. We use *cckbr1*^{+/-} as controls because the phenotype between wildtype and *cckbr1*^{+/-} were similar, which is consistent with many other KO studies. However, following the comment of the reviewer, we reanalyzed the expression of *fshb*, *lhb*, and *tshb* in the pituitaries with *cckbr1*^{+/+}, *cckbr1*^{+/-}, and *cckbr1*^{-/-} using older fish available. The results showed that *cckbr1*^{+/+} and *cckbr1*^{+/-} showed the same trend for *fshb* and *tshb*. Unfortunately, due to old age, the expression of *lhb* was lower even in wildtype and hetero fish. Since all our other data in this paper and in theory indicate that *lhb* increases with GSI, this specific data is attributed to the old fish used in this study. To prevent any misunderstanding among readers, we have chosen not to replace this data and instead highlight it in this rebuttal as evidence that wildtype and hetero fish follow the same trend.

3. AUTHORS NEED TO CLEARLY STATE THROUGHOUT THE TEXT THE SOURCE OF ALL PEPTIDE AND ANTIBODIES. ARE THE PEPTIDES FROM FISH? THE CCK ANTIBODY THEY USED IS MADE AGAINST HUMAN PROTEIN THUS THE AUTHORS SNEED TO CALL THE LABELLING “CCK-LIKE”

Response: For peptides, we have provided information about the source of the peptide (e.g. medaka CCK8). We rechecked through the manuscript and added appropriate descriptions in the revised manuscript. We indeed used cholecystokinin antibody raised against human CCK8, which differs from medaka CCK8 by only one amino acid. Due to the similarity of the peptides, we specifically labeled CCK neurons, as demonstrated by the lack of immunoreactivity in the CCK KO medaka (Supplementary Fig.6). Therefore, we believe it is appropriate to refer to the labeling as CCK-immunoreactive. However, we partially agree with the reviewer's suggestion to initially refer to it as CCK-LIKE. We have made the necessary revisions to the manuscript accordingly.

4. AUTHORS NEED TO FULLY EXPLAIN GNRH IN FISHES (3 FORMS IN GENERAL) AND THE KO-PHENOTYPE IN ZEBRAFISH.

Response: We thank the reviewer for the suggestion. We agree that explaining GnRH in fish is important. The explanation has been added to the manuscript as follows:

It is well-known that there are three paralogs of *gnrh* in vertebrates⁵⁴⁻⁵⁷, and usually one of them that is expressed in hypophysiotropic neurons is responsible for gonadotropin release (e.g. *gnrh1* in medaka; *gnrh3* in zebrafish)^{58,59}. However, in zebrafish, knocking out *gnrh3*, which is responsible for gonadotropin release, or even all of their *gnrh* genes, *gnrh2* and *gnrh3*, did not affect fertility^{11,60}. The occurrence of folliculogenesis in them can be explained by FSH-RH playing a role in regulating FSH release in the absence of GnRH, which is similar to that of medaka. For their ability of final oocyte maturation and ovulation, other compensatory factors might be taken into consideration⁶¹⁻⁶⁴. Nonetheless, the new system found in the present study may provide hints toward understanding such questions.

5. AUTHORS NEED TO EXPLAIN WHY THEY DID NOT FOLLOW-UP OIN THE FINDING OF GNRH-R2 IN THE RNASEQ ANALYSIS.

Response: The reason why we did not follow up on the GnRH-R2 in FSH cells is GnRH knockout does not decrease the FSH expression in this species (Takahashi et al., 2016), which is explained in the introduction in the original manuscript. After considering the reviewer's comment, we added a comparison of the effect of GnRH and CCK on the Ca²⁺ response of FSH cells. Although the main conclusion is that the essential regulator of FSH

is CCK but not GnRH, we do not deny the existence of GnRH regulation on FSH release. This additional experiment is shown in Supplementary Fig. 8 and explained in the main text. Also, we added a dotted arrow of GnRH to FSH in the summary figure (Fig. 6a).

Supplementary Fig. 8. [Ca²⁺]_i of FSH cells when perfused with 100 nM mdGnRH or 100 nM CCK8 peptide.

a Ca²⁺ imaging of FSH cells applied with 100 nM mdGnRH or 100 nM CCK8

b The response of FSH cells to CCK8 peptide is significantly greater than that to mdGnRH at 100 nM ($n=5$). * $P<0.05$.

6. ELIMINATE THE DATA ON THE JAPANESE EEL, IT IS NOT FULLY EXPLORED AND AGAIN THE ANTIBODIES ARE NOT AGAINST THE PEPTIDE OF THIS ANIMAL.

Response: Thank you for the suggestion. To highlight the significance of including data on the Japanese eel, we added additional data on the double in situ hybridization, but not immunohistology, of eel pituitary labeling *lhb* and *cckbr1* in Supplementary Figure 13. We made this addition not only because it was suggested by the other reviewer, but also because we believe it further supports the evolutionary aspects and potential conservation of this system in teleosts.

TEXT:

Therefore, to challenge the canonical “solo GnRH model,” we aimed to identify the other gonadotropin regulator,

CHANGE WORDING: replace challenge with investigate

Response: In accordance with the comment and due to the word limit of the abstract, we deleted this whole sentence.

In the present study, to challenge the current “solo GnRH model,” we aimed to identify the FSH-RH.

CHANGE WORDING: replace challenge with investigate

Response: In accordance with the comment, we reconsidered the wording. As *investigating the current “solo GnRH model” (in medaka)* might be misleading, we changed it to “reevaluate”.

However, this once-established consensus has been challenged in vertebrates other than 55 mammals. Intriguingly, it has been reported that GnRH knockout (KO) does not affect FSH

56 function in model teleosts such as medaka and zebrafish 10,11, which implies that GnRH may not 57 be the primary regulator of FSH release, at least in teleosts.

AUTHORS NEED TO MORE CLEARLY EXPLAIN THESE PAPERS, AS THE SENTENCE IS WRITTEN IT SOUNDS LIKE THE KO ONLY AFFECTS FSH FUNCTION, THIS IS NOT TRUE: THE ANIMALS ARE FULLY FERTILE. ALSO EXPLAIN THE DIFFERENT FORMS OF GNRH (MEDAKA HAVE THREE, ZEBRAFISH HAVE 2)

Response: We thank the insightful comment. In the revised version, we believe we have resolved this point while addressing a similar issue mentioned in the comment, General Comment #4.

75 Among the metabolic receptors expressed, we found that cholecystokinin B receptor 1 (cckbr1)76 had the highest expression (Supplementary Table 1).

WHY DO AUTHORS DISCOUNT GNRH-R2, WHICH IS THE SECOND HIGHEST EXPRESSION?

Response: This comment has been addressed in our comments to General Comment #5.

Thus, we 105 proved that *Cckbr1*, which is expressed in FSH cells, is essential for the normal function of FSH

AGAIN “PROVED” IS A VERY STRONG WORD,

Response: Thank you for the comment. We agree that “proved” is a very strong word, especially with the results that we had. However, we added a rescue experiment to further strengthen the concept that *Cckbr1* in FSH cells is essential for the function of FSH (Fig.2).

Fig. 2. FSH cell-specific rescue of *Cckbr1* in *cckbr1* KO medaka recovered *fshb* expression and fertility in females.

a Immunohistochemistry of the pituitary of wildtype and *cckbr1*^{-/-} medaka with the rescue transgene (*fshb*:*Cckbr1*-FLAG). The pituitary is labeled with transgenically introduced *Cckbr1* (FLAG-tagged, green) and intrinsic FSH β (magenta). FSH cells are successfully forced to express FLAG-tagged *Cckbr1*. Note that the pituitary of wildtype medaka shows normal expression of FSH but shows no immunoreactivity to FLAG. **b** Ovarian size of *cckbr1*^{-/-} females with or without the rescue transgene. The data are mean \pm SEM. **c-e** Expression of *fshb* (c), *lhb* (d), and *tshb* (e) in the pituitary of each *cckbr1*^{-/-} with or without the rescue transgene. The data are mean \pm SEM. *** P <0.001, ** P <0.01, N.S. not significant.

Hence, the result is described in the revised manuscript as follows:

The essentiality of *Cckbr1* in FSH cells was further examined by rescuing *cckbr1*

specifically in FSH cells in *cckbr1*^{-/-} medaka. Here, we generated a transgenic medaka harboring a rescue transgene containing C-terminal FLAG-tagged Cckbr1 coding sequence under the promoter of *fshb*, which expresses Cckbr1 specifically in FSH cells (Fig.2a). After crossing with *cckbr* KO medaka, the effect of the rescue transgene was examined. *cckbr1*^{-/-} medaka with the rescue transgene showed spawning unlike their siblings without the transgene. To further assess the effect of the rescue transgene, the ovary sizes of *cckbr1*^{-/-} samples with and without the rescue transgene were examined. It was revealed that the ovary size of the medaka ($n = 6$) with the transgene showed significant increase in ovarian size compared to the sample without the transgene (Fig. 2b). Additionally, the mRNA expression of *fshb*, *lhb*, and *tshb* in the pituitary was analyzed using qRT-PCR. As implied from the ovary size, the *fshb* mRNA expression of the pituitary with the rescue transgene showed a significant increase compared to those without the transgene (Fig. 2c). The *lhb* expression of the rescued KO also had a significant increase, which is explained by the secondary effect of the rescued FSH function (Fig. 2d). The *tshb* expression did not change in either group (Fig. 2e). Thus, Cckbr1, which is expressed in FSH cells, is proven to be crucial for the FSH regulation.

Differ among all genotypes (Fig. 4d, e). The fact that only the double KO was associated with a

181 severe phenotype suggests that *ccka* and *cckb* have redundancy in their function of FSH

CROSSES ARE NEEDED HERE TO CONFIRM FERTILITY DEFECTS

Response: We thank you for the suggestion. We agree that crosses are necessary to confirm the fertility defect in the KO medaka. Therefore, we crossed the female *ccka/cckb* KO genotypes with fertile male d-rR wildtype and counted the eggs. Throughout the course of the experiment, only *ccka/cckb* DKO began to spawn one month later than the other genotypes. The result is shown in Supplementary Figure 11.

Supplementary Fig. 11. Eggs spawned from *ccka/cckb* medaka of each genotype
a The number of eggs spawned from female of each genotype paired with wild type male ($n = 4$). The data are mean \pm SEM. **b** The number of females that spawned eggs in each genotype. *ccka*^{-/-};*cckb*^{-/-} medaka began to lay eggs about one month after the other genotypes.

The main text and as follows:

Interestingly, some of the double knockouts started spawning about one month after the wildtype started to spawn (Supplementary Fig. 11) even with completely reduced *fshb* expression and significantly smaller ovary ($n = 4$, Supplementary Fig. 12).

183 hypophysiotropic neurons (Fig. 2). Interestingly, some of the double knockouts started spawning 184 one month after the wild type started to spawn even with completely reduced *fshb* expression and 185 significantly smaller ovary (n = 4), which can be explained by some compensatory mechanism 186 that involves LH action on FSH receptor (Supplementary Fig. 8). These results indicate that 187 cholecystokinin is the primary factor responsible for the expression and release of FSH, which 188 should be referred to as the FSH-RH.

THIS IS A WORRYING OBSERVATION AND IS WHY ALL GENOTYPES NEED TO BE CROSSED AND THE NUMBER OF EMBRYOS SCORED, AND THESE DATA INCLUDED IN THE MANUSCRIPT

Response: We thank the reviewer for the suggestion. In accordance with the reviewer's comment, we counted the number of eggs spawned from *ccka/cckb* double knockout females when crossed with fertile male wildtype medaka. It was confirmed that *ccka/cckb* double knockout females began to spawn eggs ~3 months post-hatch, which represents a delay of approximately 1 month compared to other genotypes that spawned eggs 2 months post-hatch (Supplementary Fig.11, shown in the previous comment). We have also performed RT-PCR for each genotype, which revealed a significant reduction in *fshb* expression, while *lhb* expression was not significantly reduced compared to the other genotype. Given the activity of LH on FSH receptor, we can explain the occurrence of this delayed spawning as a result of a compensatory mechanism of LH. This explanation was added to the manuscript as follows:

This can be explained by the compensatory mechanisms involving LH as follows. First, a small amount of FSH is released by another ligand of *Cckbr1*, perhaps Gastrin, which slowly stimulates folliculogenesis. This causes an increase in serum estrogen, which should induce LH secretion²⁴. Here, in medaka, because LH can also activate FSH receptor³⁸, LH can regulate the ovulatory cycle in an FSH-independent manner once LH secretion is activated. Despite the occurrence of this delayed spawning, the overall results consistently indicate a severe deficiency of the FSH system in *ccka/cckb* double KO. Therefore, cholecystokinin is the primary factor responsible for the expression and release of FSH, which should be referred to as the FSH-RH.

METHODS

To label the CCK-expressing cells, we used an anti-cholecystokinin (26-33) antibody raised in

316 rabbit (C2581, Sigma-Aldrich, St. Louis, MO; 1:5,000).

THIS IS A HUMAN SPECIFIC ANTIBODY: AUTHORS NEED TO CHANGE THE WORDING THROUGHOUT THE TEXT OF THE MANUSCRIPT TO “CCK-LIKE”

Response: This comment has been addressed in our comments to General Comment #3.

Double in situ hybridization was visualized by TSA plus biotin

301 followed by ABC Elite kit (Vector Laboratories, Burlingame, CA) and Streptavidin, Alexa

302 Fluor 488 (Green; Thermo Fisher, Waltham, MA), and TSA plus Cy3 (Red; Akoya Bioscience, 303 Marlborough, MA). Note that the first peroxidase label on the fluorescein probe was completely 304 quenched with 3% H₂O₂ for 40 mins before the second antibody for the DIG probe was labeled.

THIS SECTION NEEDS TO BE RE-WRITTEN. AS IT APPEARS HERE THE TWO DIFFERENT FLUORESCENT FLUOROCHROMES IN THE ISH ARE UNCLEAR: PLEASE STATE CLEARLY WHICH PROBES WERE LABELED WITH DIG AND WHICH WERE LABELLED WITH FLUORESCENT LABELLED PROBES. DID THE AUTHORS USE ANTI-DIG AND ANTI-FLUORESCEIN? IT LOOKS LIKE THE AUTHORS COPIED THE SECTION FROM IMMUNOCYTOCHEMISTRY FOR ANTIBODIES.

Response: We thank you for the comment. We have indeed used different fluorochromes in the ISH. The specific fluorochromes and other reagents used for each detection are detailed in the revised manuscript as follows:

In double *in situ* hybridization the fluorescein-labeled probe (*cckbr1* or *ccka*) was labeled by Anti-Fluorescein-HRP Conjugate (Perkin Elmer, Waltham, MA). The sections were subjected to TSA reaction using TSA plus biotin (Akoya Bioscience, Marlborough, MA). The biotin signal was visualized by Streptavidin, Alexa Fluor 488 (Green; Thermo Fisher, Waltham, MA) after signal amplification with ABC elite kit (Vector Laboratories, Burlingame, CA). Then the first peroxidase activity label on the fluorescein probe was completely quenched with 3% H₂O₂ for 40 minutes before the application of an antibody to the DIG probe. The expression of *fshb*, *lhb*, or *cckb* was detected using DIG-labeled probes. The hybridized DIG-labeled probes were further detected by anti-digoxigenin-POD Fab fragments (Sigma Aldrich). The POD activity was then visualized by TSA plus

Cy3 (Red; Akoya Bioscience).

- SUPP FIGURE 11 DIAGRAM: EXPLAIN WHAT THE GREEN AND BLUE COLORS MEAN

Response: We thank the reviewer's comments. The green color indicates the untranslated regions of the exon while the blue indicates the coding regions of the exon. We made the schematics of the gene structures less ambiguous. Additionally, in accordance with the reviewer's comment, the alignment and schematics of the KOs of each genotype have been split into 3 different supplementary figures (Supplementary Figure 15, 16, and 17).

- PLEASE INCLUDE ALIGNMENT OF SEQUENCES FROM THE WILD TYPE AND MUTANT GENES (REGION THAT IS DELETED), ESPECIALLY IMPORTANT FOR THE CCKBR1 SEQUENCE.

Response: Thank you for the insightful comment. We agree that we needed more data regarding the sequence not only of *cckbr1* but also of *ccka* and *cckb*. Therefore, we added supplementary figures indicating the alignment sequences (Supplementary Fig. 15, 16, 17).

• PLEASE GIVE A MORE THOROUGH DESCRIPTION OF THE MUTANTS: DATA NEED TO BE PRESENT FOR CROSSES OF THE FISH TO SHOW THAT THEY ARE TRULY AFFECTED BY THE MUTATION. WHAT IF THE OVARIES ARE REDUCED IN SIZE BUT FISH STILL CAN LAY A FEW EGGS.

Response: We thank the reviewer for the suggestion. In response to the comment, we performed additional experiments to examine the female fertility of each genotype by crossing them with wildtype males. Our findings indicate that all *cckbr1* KO males displayed normal spermatogenesis and fertility, similar to what has been previously reported for *fshb* KO. Therefore, we believe that the examination of *cckbr1* KO females with wild-type males sufficiently demonstrates the significance of this system. We have added Supplementary data that present the number of individuals that spawned eggs, as well as the average number of eggs spawned in *cckbr1* KOs (Supplementary Fig. 3) and *ccka/b* KOs (Supplementary Fig.11).

Supplementary Fig. 3. Eggs spawned from *cckbr1* medaka of each genotype.

a The number of eggs spawned from female of each genotype paired with wild type male ($n = 5$). The data are mean \pm SEM. **b** The number of females that spawned eggs in each genotype when paired with a wild type male. No eggs were spawned from *cckbr1*^{-/-} while *cckbr1*^{+/+} and *cckbr1*^{+/-} spawned.

Supplementary Fig. 11. Eggs spawned from *ccka/cckb* medaka of each genotype

a The number of eggs spawned from female of each genotype paired with wild type male ($n = 4$). The data are mean \pm SEM. **b** The number of females that spawned eggs in each genotype. *ccka*^{-/-};*cckb*^{-/-} medaka began to lay eggs about one month after the other genotypes.

During the duration of the experiment, *cckbr1* KOs did not spawn eggs, while *cckbr1*^{+/+}

and *cckbr1*^{+/-} spawned eggs around the same time. We conducted a similar procedure with *ccka/cckb* KOs. We have already explained in the previous comment and discussed it in the main text.

- ARE THEY FERTILE AS HETEROZYGOTES?

Response: The size of the ovary, as well as the gonadosomatic index (GSI), was not significantly different between wildtype and *cckbr1* hetero KO (Fig. 1d,e). Further test was conducted by crossing *cckbr1* KO genotypes with fertile wildtype males. The eggs spawned between wildtype and *cckbr1* hetero were not different. The data has been added to Supplementary Fig.3 (shown in the previous comment). Also, this fact has been described in the main text as follows:

No spawning was observed in *cckbr1*^{-/-} while fertilized eggs were observed in *cckbr1*^{+/+} and *cckbr1*^{+/-} fish. The number of eggs spawned in each genotype during 60 days post-hatch (dph) to 105 dph showed that *cckbr1*^{+/+} and *cckbr1*^{+/-} started to spawn eggs around 70 dph and the number of eggs peaked around 90 dph, whereas *cckbr1*^{-/-} did not spawn any eggs by 105 dph. (Supplementary Fig. 3a,b).

- WHAT HAPPENS WHEN YOU CROSS THESE KO FISH?

THESE DATA NEED TO BE SHOWN FOR EACH KO (SEE OTHER EXAMPLES IN THE LITERATURE, LIKE ZOHAR LAB)

Response: We thank you for the insightful question. In accordance with the reviewer's comment, we crossed these female KO fishes with fully fertile wildtype males and counted the eggs spawned. The results showed that the KO females did not lay eggs, which is consistent with the fact that *fshb* KO females but males are fertile. Furthermore, based on the morphology of gonads and all *in vitro* analyses, we determined that the problem is related to the folliculogenesis process in the gonad. As already explained in the previous comments, these data are shown in Supplementary Fig.3 and Supplementary Fig.11 and explained in the main text and the previous comment.

ALSO IT APPEARS THE AUTHORS INJECTED ON A SINGLE GUIDE RNA FOR EACH GENE. IF THIS IS TRUE THEY NEED TO EXPLAIN WHY THEY GENERATED MUTANTS WITH SUCH LARGE DELETIONS: CCKA 743-bp deletion, CCKB 210-bp deletion

Response: We thank the reviewer's comment. It is our mistake in the original manuscript. Since we used two guide RNAs for *ccka* and *cckb* each. We modified it in the

revised manuscript as follows:

Target sequences of the CRIPSR RNA including PAM are as follows. *Cckbr1*, AAGCGTGGACGGGTTACGCAGG; *ccka*, TGACGCGTGTGATTGGTTAGTGG and ACCTGGGATGGATGGACTTTGGG; *cckb*, GGAGTGCTGGCCCTCATCTGAGG and GCAGCTGAAAGACCTTCCCGGGG.

FIGURES:

Authors make statements about fibers when the figures are of too low magnification to draw conclusions (see Supp Fig 5)

Response: In accordance with the reviewer's comment, we added the picture in higher magnification. The revised figures is as follows (Supplementary Fig.6 in the revised version).

Supplementary Fig. 6. *ccka* and *cckb* knockout fish do not have CCK-immunoreactive cell bodies and fibers.

a, Immunohistochemistry of *ccka* and *cckb* double knockout medaka, using CCK antibody. Cell body and fibers were observed in the pituitary, preoptic area (POA), and nucleus ventralis tuberis (NVT) of *ccka*^{+/+};*cckb*^{+/+} medaka. **b**, No cell body or fibers were observed in *ccka*^{-/-};*cckb*^{-/-} medaka. The bottom half of the figures are in higher magnification. Scale bars, 50 μ m.

Supp Fig 7 has a clear sex bias in the male versus female data: why is there no assay of ovary weight?

Response: The corresponding data of females (ovary weight) is shown in the main figure so that there is no data in the supplementary figure in the original manuscript. To avoid confusion, we moved the testes size from Supplementary Figure 10 to the main figure (Figure 5) and arranged them uniformly for both sexes.

Supp fig 8: confusing: if *ccka/b* double knock shows no phenotype (See Panels in a), how can authors conclude it controls FSH (which controls EARLY gonad development)?

Response: This is a misunderstanding of the reviewer. The *ccka/b* double KO has a clear phenotype in their *fshb* expression (Fig. 5d, Supplementary Fig. 12c of the revised manuscript).

Fig. 5d

Supplementary Fig.12c

Even though some of these KO fish begin spawning 1 month later compared to other genotypes (Supplemental Fig. 11, see previous comments), they showed a drastically decreased *fshb* expression (~99%; Supplemental Fig. 12c), which suggests compensation by LH because FSH-receptor is also activated by LH. This explanation has been added in the revised main text and mentioned in the previous comment.

Supp figure 9 is expression in the Japanese eel (?)

Response: As described in the figure legend of Supplementary Figure 9 in the original manuscript, this is the data of Japanese eel. In the revised version, we confirmed it indicated twice in the legend to avoid oversight by the readers.

References

- Karigo, T. et al. Whole brain-pituitary in vitro preparation of the transgenic medaka (*Oryzias latipes*) as a tool for analyzing the differential regulatory mechanisms of LH and FSH release. *Endocrinology* **155**, 536-547 (2014).
- Hoffmann, P. et al., Comparison of clearance and metabolism of infused cholecystokinins 8 and 58 in dogs. *Gastroenterology* **105**, 1732-1736 (1993).
- Kanda, S., Okubo, K. & Oka, Y. Differential regulation of the luteinizing hormone genes in teleosts and tetrapods due to their distinct genomic environments - Insights into gonadotropin beta subunit evolution. *Gen Comp Endocrinol* **173**, 253-258 (2011).

Reviewers' comments:

Reviewer #1 (Remarks to the Author):

The authors have adequately resolved most of the concerns I raised. However, further elaboration on the distinction between the dual and single GnRH model is warranted. I appreciate the authors' effort in addressing my previous comment (Point 2#) by conducting additional Ca²⁺ imaging experiments on FSH cells, which indeed displayed a weak response to GnRH and a strong response to CCK8. This outcome is promising. To strengthen their argument for the dual GnRH model, I suggest the authors also investigate the responses of LH cells to both peptides in a similar experimental setup. If the dual GnRH model is true, one would anticipate that the FSH cells respond strongly to CCK8 but weakly to GnRH, while LH cells show opposite reactions, with a strong response to GnRH but not to CCK8. Such experiment would provide a more comprehensive understanding of the peptide-specific responses in these cell types.

Reviewer #2 (Remarks to the Author):

STILL DO NOT UNDERSTAND WHY THERE IS NOT A BIGGER DIFFERENCE IN GONAD SIZE (FEMALES SHOULD SMALLER THAN MALES) IF CCK IS DRIVING THE FSH PATHWAY (FIG 1) ALSO WHY DOES THE RECEPTOR HAVE SUCH A BIG EFFECT (INCLUDING IN MALES) BUT THEY HAVE TO KNOCK OUT BOTH LIGANDS TO HAVE A PHENOTYPE?

MALES HAVE SMALLER GONADS BUT NORMAL SPERMATOGENESIS...AUTHORS NEVER TELL OF THE FERTILITY (FERTILIZATION) RATE SO HOW DO WE KNOW THE SPERM ARE NORMAL?

AUTHORS NEED TO RESCUE *cckbr1* ^{-/-} MALES WHOSE GONADS ARE NOTABLY SMALLER AS SHOWN IN FIGURE 1

WHY DID MALES RESPOND?

In FSH cells, 1 μ M CCK8 induced a rapid and strong intracellular Ca²⁺ increase, which triggers hormonal release from the FSH cells (Fig. 4a). This effect was also observed in males (Supplementary Fig. 7).

PLEASE ADDRESS WHY MALES WERE NOT TESTED:

Although mdGnRH (100nM) induced a Ca²⁺ response in FSH cells as suggested in a previous study, this response was significantly lower than that induced by CCK8 at the same concentration (n = 5, Supplementary Fig. 8a, b). Therefore, we suggest here that CCK can be the primary hypophysiotropic factor to regulate FSH release.

THIS SHOULD BE FURTHER INVESTIGATED

Interestingly, although a single KO of *ccka* or *cckb* resulted in a normal phenotype, the double KOs showed a severe change in phenotype

AGAIN, WHAT IS HAPPENING IN THE MALES?

The overall phenotype of the double KO was 224 similar to that of the cckbr1 KO. In both females and males, the gonadal size of ccka/cckb 225 double KO was drastically decreased

THIS IS VERY IMPORTANT: AREN'T THE AUTHORS CURIOUS OR PERHAPS WORRIED ABOUT THIS EFFECT?

Interestingly, some of the 232 double knockouts started spawning about one month after the wildtype started to spawn 233 (Supplementary Fig. 11) even with completely reduced fshb expression and significantly smaller 234 ovary (n = 4, Supplementary Fig. 12). Despite the occurrence of this delayed spawning, the overall results 240 consistently indicate a severe deficiency of the FSH system in ccka/cckb double KO.

THIS IS TOO STRONG A STATEMENT, AND BECAUSE THE AUTHORS HAVE NOT RESOLVED THE EFFECTS IN MALES, NOR EXPLAINED THE RESTORED FECUNDITY THEY ARE POTENTIALLY MISSING IMPORTANT ASPECTS OF THE MECHANISM

Therefore, 241 cholecystokinin is the primary factor responsible for the expression and release of FSH, which 242 should be referred to as the FSH-RH.

Point-by-point responses to the reviewers' comments.

We found the Referees' comments very constructive and helpful for us in revising the manuscript. We have studied your comments very carefully and have made the necessary emendations.

The point-by-point responses to the reviewers' comments are listed below. The paragraphs written in blue are the comments from the reviewers, and our response follows in black text. In the revised manuscript, the modified texts are highlighted in yellow.

Reviewer #1 (Remarks to the Author):

The authors have adequately resolved most of the concerns I raised. However, further elaboration on the distinction between the dual and single GnRH model is warranted. I appreciate the authors' effort in addressing my previous comment (Point 2#) by conducting additional Ca²⁺ imaging experiments on FSH cells, which indeed displayed a weak response to GnRH and a strong response to CCK8. This outcome is promising. To strengthen their argument for the dual GnRH model, I suggest the authors also investigate the responses of LH cells to both peptides in a similar experimental setup. If the dual GnRH model is true, one would anticipate that the FSH cells respond strongly to CCK8 but weakly to GnRH, while LH cells show opposite reactions, with a strong response to GnRH but not to CCK8. Such experiment would provide a more comprehensive understanding of the peptide-specific responses in these cell types.

As suggested by the reviewer, we performed an additional Ca²⁺ imaging analysis in the pituitary of *lh:IP* transgenic medaka perfused with 100 nM of mdGnRH or CCK8. As expected, mdGnRH induced a Ca²⁺ response in LH cells. On the other hand, no Ca²⁺ response was observed upon perfusion with CCK8 at the same concentration ($n = 5$, Supplementary Fig. 8c, d). These results indicate that CCK exclusively regulates FSH release while GnRH is the primary regulator for LH release.

Accordingly, we revised the manuscript as follows (Lines 211-214):

“Also, as reported previously, mdGnRH induced a large [Ca²⁺]_i increase in LH cells while CCK8 showed no effect on LH cells ($n = 5$, Supplementary Fig. 8c, d). Therefore, we suggest here that CCK can be the primary hypophysiotropic factor to regulate FSH release, while GnRH is the primary factor to regulate LH release.”

Reviewer #2 (Remarks to the Author):

STILL DO NOT UNDERSTAND WHY THERE IS NOT A BIGGER DIFFERENCE IN GONAD SIZE (FEMALES SHOULD SMALLER THAN MALES) IF CCK IS DRIVING THE FSH PATHWAY (FIG 1)

AGAIN, WHAT IS HAPPENING IN THE MALES?

The overall phenotype of the double KO was 224

similar to that of the *cckbr1* KO. In both females and males, the gonadal size of *ccka/cckb* 225 double KO was drastically decreased

These comments raise concerns about why *cckbr1* KO and *ccka/cckb* double KO result in smaller testes in males. However, the phenotypes of KO males showing smaller testes are reasonable because FSH signaling is important for testicular functions. This might be due to the reviewer's incorrect belief that FSH has no function in males.

This incorrect belief may have originated from previous studies demonstrating that the KO of FSH (*fshb*) or FSH-receptor in medaka and zebrafish leads to abnormal oogenesis in females, while spermatogenesis remains functioning in males (Takahashi et al., 2016; Murozumi et al., 2014; Zhang et al., 2015; Chu et al., 2015). These studies concentrated on females and did not analyze male gonads in detail because females showed a drastic phenotype. However, male FSH-receptor KO medaka have been reported to show a reduced testis weight (Kitano et al., 2022, Fig.1A; not statistically significant due to small sample size). In addition, male *fshb* KO medaka exhibit reduced gonad size (our unpublished observation). These observations that disruption of FSH-RH or FSH attenuates testis size are reasonable because FSH in males is involved in spermatogonial proliferation as well as steroidogenesis (Schulz et al., 2010; Nóbrega et al., 2015; Sambroni et al., 2013). Thus, these comments are based on misbelief, and the correct understanding of the *fshb* KO phenotype is that “*fshb* KO males are fertile but they show reduced testis size”. Therefore, our results that *cckbr1* KO and *ccka/cckb* double KO males showed fertility and reduced testis size are reasonable.

To avoid such misunderstanding for the readers, we added the description of the current understanding of the FSH functions in males in the revised manuscript as follows (Lines 123-128):

“Although there are no previous studies that have examined the GSI of *fshb* KO males in medaka, FSH is generally considered to stimulate spermatogenesis including spermatogonial proliferation in teleosts^{28,27,29}. Therefore, the phenotype of *cckbr1*^{-/-} males that show reduced GSI (Fig. 1c) can be also explained by their reduction in FSH secretion (Fig. 1k). Thus, *Cckbr1* was shown to have a pivotal role in FSH secretion in both sexes.”

ALSO WHY DOES THE RECEPTOR HAVE SUCH A BIG EFFECT (INCLUDING IN MALES) BUT THEY HAVE TO KNOCK OUT BOTH LIGANDS TO HAVE A PHENOTYPE?

We have clearly described the reason for this point in both the original and revised manuscripts. The key point is that the ligands *ccka* and *cckb* are co-expressed in pituitary-projecting neurons. Therefore, a single knockout for each gene is insufficient

to disrupt this system. The following sentence is the explanation provided in the revised manuscript (Lines 248-251):

“The fact that only the double KO was associated with a severe phenotype suggests that *ccka* and *cckb* have redundant functions in FSH regulation, which is also consistent with the fact that *ccka* and *cckb* co-localize in the NVT hypophysiotropic neurons (Fig. 3c).”

MALES HAVE SMALLER GONADS BUT NORMAL SPERMATOGENESIS...AUTHORS NEVER TELL OF THE FERTILITY (FERTILIZATION) RATE SO HOW DO WE KNOW THE SPERM ARE NORMAL?

Given that *fshb* KO males can reproduce with smaller gonads, it is natural to assume that *cckbr1* (FSH-RH receptor) KO and *ccka/cckb* double KO males are also fertile, albeit with smaller gonads. Therefore, in the original manuscript, we did not explicitly state that males are fertile. Additionally, we did not include this information in the previously revised manuscript because there were no specific comments requesting information about the male phenotype during the first round of review.

In the present revised manuscript, to clarify the idea above, we performed an additional experiment to analyze the average number of spawned eggs, the average number of fertilized eggs, and the fertilization rate of *cckbr1* KO or wildtype female medaka when paired with *cckbr1* KO or wildtype male medaka ($n = 4$). As expected, no *cckbr1* KO females spawned when paired with the wildtype males, while *cckbr1* KO males spawned when paired with the wildtype females (Supplementary Fig. 3a-c). We also examined the fertility of *ccka/cckb* KO females and males. Among the pairs examined, no *ccka/cckb* double KO females spawned, while *ccka/cckb* double KO males showed fertility (Supplementary Fig. 11a-c). These results demonstrate that *cckbr1* and its ligands *ccka* and *cckb* are dispensable for male fertility, albeit with smaller testis, which is consistent with the FSH KO phenotype.

AUTHORS NEED TO RESCUE *cckbr1*^{-/-} MALES WHOSE GONADS ARE NOTABLY SMALLER AS SHOWN IN FIGURE 1

In accordance with the reviewer's comment, we performed an additional experiment with or without the rescue transgene in male *cckbr1*^{-/-}. Similar to females, the reintroduction of *cckbr1* in FSH cells led to a significant increase in testis size and *fshb* expression.

Accordingly, we revised the manuscript as follows (Lines 146-159):

“To further assess the effect of the rescue transgene, the ovary and testis sizes of *cckbr1*^{-/-} medaka with or without the rescue transgene were examined. Our findings revealed a significant increase in gonadal size for both the ovary ($n = 6$) and testis ($n = 4$) when the transgene was present, compared to their siblings without the transgene (Fig. 2b, 2f). We also analyzed the mRNA expression of *fshb*, *lhb*, and *tshb* in the pituitary using qRT-PCR. Consistent with the ovary and testis sizes, the pituitary with the rescue transgene showed a significant increase in the *fshb* mRNA expression compared to those without the transgene, in both females and males (Fig. 2c, 2g). In females, the *lhb* expression of the rescued KO showed a significant increase, which can be explained by the secondary effect of the rescued FSH function (Fig. 2d). In contrast, in males, no significant change in *lhb* expression was observed (Fig. 2h), which is consistent with the KO result that showed no reduction in *lhb* expression in males (Fig. 11). The *tshb* expression did not change in either group, both in females and males (Fig. 2e, 2i). Thus, *Cckbr1*, which is expressed in FSH cells, has been proven to be crucial for FSH regulation.”

PLEASE ADDRESS WHY MALES WERE NOT TESTED:

Although mdGnRH (100nM) 193 induced a Ca²⁺ response in FSH cells as suggested in a previous study 30, this response was 194 significantly lower than that induced by CCK8 at the same concentration (n = 5, Supplementary 195 Fig. 8a, b). Therefore, we suggest here that CCK can be the primary hypophysiotropic factor to 196 regulate FSH release.

This is because these actions have already been shown to have no sexual dimorphism in the present (for CCK) and in a previous study (GnRH). We have already shown that the FSH cells perfused with CCK8 showed similar responses in females (Fig. 4a) and males (Supplementary Fig. 7). Also, a previous study clearly stated that mdGnRH increases Ca²⁺ levels similarly in males and females (Karigo, et al., 2014).

To address this fact more clearly, we have added the following sentence in the main text (Lines 214-216):

“Note that this conclusion applies regardless of sex, as evidenced in the present study (Fig. 4a and Supplementary Fig.7) as well as in a previous study³³.”

If further evidence is required, we are willing to perform additional similar experiments using males.

WHY DID MALES RESPOND?

In FSH cells, 1uM CCK8 induced a rapid and strong intracellular Ca²⁺ 186 increase, which triggers hormonal release from the FSH cells (Fig. 4a). This effect was also 187 observed in males (Supplementary Fig. 7).

These questions might be based on the incorrect belief of Reviewer #2 that FSH-RH and FSH are female-specific (Please see Reviewer #2's first comment). However, since FSH is functional in both females and males (Schulz et al., 2010; Nóbrega et al., 2015; Sambroni et al., 2013), it is reasonable to assume that its regulator, the FSH-RH system, is present in both sexes. In fact, in the present study, all our results indicate that the FSH-RH system plays an important role in FSH secretion in both sexes. Specifically, both female and male *cckbr1* KO fish showed a ~99% decrease in *fshb* expression (Fig. 1h and k), indicating that the FSH-RH system is essential for FSH secretion in both sexes. Additionally, it is worth noting that GnRH has also been reported to stimulate LH release in both females and males (Karigo et al., 2014). Therefore, our finding that CCK8 increased intracellular Ca²⁺ in FSH cells in males is reasonable, as it is consistent with our results and the current understanding of FSH function.

To prevent readers from misunderstanding, we have described the importance of FSH function on gonadal function regardless of sex in the revised manuscript as follows (Lines 123-128):

“Although there are no previous studies that have examined the GSI of *fshb* KO males in medaka, FSH is generally considered to stimulate spermatogenesis including spermatogonial proliferation in teleosts^{28,27,29}. Therefore, the phenotype of *cckbr1*^{-/-} males that show reduced GSI (Fig.1c) can be also explained by their reduction in FSH secretion (Fig. 1k). Thus, *Cckbr1* was shown to have a pivotal role in FSH secretion in both sexes.”

THIS SHOULD BE FURTHER INVESTIGATED

Interestingly, although a single KO of 222 *ccka* or *cckb* resulted in a normal phenotype, the double KOs showed a severe change in 223 phenotype

These results are reasonable because *ccka* and *cckb* are co-expressed in the same neurons. A severe phenotype should only be observed when both of these genes are knocked out.

This explanation is provided in both the original and revised versions of the manuscript (Lines 248-251):

“The fact that only the double KO was associated with a severe phenotype suggests that *ccka* and *cckb* have redundant functions in FSH regulation, which is also consistent with the fact that *ccka* and *cckb* co-localize in the NVT hypophysiotropic neurons (Fig. 3c).”

THIS IS VERY IMPORTANT: AREN'T THE AUTHORS CURIOUS OR PERHAPS WORRIED ABOUT THIS EFFECT?

Interestingly, some of the 232 double knockouts started spawning about one month after the wildtype started to spawn 233 (Supplementary Fig. 11) even with completely reduced *fshb* expression and significantly smaller 234 ovary (n = 4, Supplementary Fig. 12).

Despite the occurrence of this delayed spawning, the overall results 240 consistently indicate a severe deficiency of the FSH system in *ccka/cckb* double KO.

This might be due to the oversight by the reviewer. We have already described reasonable explanations in the previous revision, in the Result and Discussion sections.

The most important facts explained are as follows:

1. This delayed spawning is due to the deficiency of FSH function in *ccka/cckb* double KO females (less than 7% *fshb* expression of wild type)
2. They spawn with this drastically attenuated FSH function, probably due to the compensatory effect by LH because LH can also activate FSH receptor (Ogiwara et al., 2013).

The <7% *fshb* expression observed in *ccka/cckb* double KO females can be attributed to Gastrin, which is another ligand for *Cckbr1* and is released from the gut, but not from the brain (Supplementary Fig. 5). Despite the existence of such a sub-regulator, it is important to note that >93% of *fshb* expression is explained by *Ccka* and *Cckb*. These results and explanations indicate that *Ccka* and *Cckb* are the “primary regulators” of FSH secretion.

To prevent readers from misunderstanding, we emphasized these explanations as follows (added texts are shown in red) in lines 256-266 and lines 331-340:

“Interestingly, some of the double knockouts started spawning about one month after the wildtype started to spawn (Supplementary Fig. 11d, e) even with completely reduced *fshb* expression and significantly smaller ovary (n = 4, Supplementary Fig. 12). This can be explained by the compensatory mechanisms involving LH as follows. First, a small amount of FSH (with *fshb* expression of <7% of wildtype) is released by another ligand of *Cckbr1*, perhaps Gastrin, which slowly stimulates folliculogenesis. This causes an increase in serum estrogen, which should induce LH secretion in a positive feedback manner²⁴. Here, in medaka, because LH can also activate FSH receptor³⁸, LH can regulate the ovulatory cycle in an FSH-independent manner once LH secretion is activated. Despite the occurrence of this delayed spawning, the overall results consistently indicate a severe deficiency of the FSH system in *ccka/cckb* double KO.”

“The *ccka/cckb* double homozygote KO reduced the expression level of *fshb* to <7% of *fshb* expressed compared to the double heterozygote KO of the same batch (Fig. 5), whereas *cckbr1* homozygote KO resulted in ~0.5% of the heterozygote KO (Fig. 1). This difference suggests the existence of a stimulator of *Cckbr1* other than CCK. A candidate for the compensatory factor is Gastrin, which has comparable biological activity to the CCK receptor (Fig. 3a). In PCR analysis, *gast* mRNA was detected in the gut rather than the hypothalamus (Supplementary Fig. 5). Therefore, this possible stimulator may come from the gut, which implies gut-derived CCK and Gastrin affect FSH release. However, it should be noted that this effect should be much weaker than

that from hypophysiotropic CCK neurons, which directly innervate to the FSH cells (Fig. 3e, f).”

THIS IS TOO STRONG A STATEMENT, AND BECAUSE THE AUTHORS HAVE NOT RESOLVED THE EFFECTS IN MALES, NOR EXPLAINED THE RESTORED FECUNDITY THEY ARE POTENTIALLY MISSING IMPORTANT ASPECTS OF THE MECHANISM

Therefore, 241 cholecystokinin is the primary factor responsible for the expression and release of FSH, which 242 should be referred to as the FSH-RH.

As explained above, the criticism toward the male results is based on a misunderstanding of the reviewer and oversights of our explanation for the restored fecundity. All the data and supporting evidence support our conclusion that cholecystokinin is the “primary factor” responsible for the expression and release of FSH, which can be referred to as FSH-RH.

REVIEWER COMMENTS

Reviewer #1 (Remarks to the Author):

The authors have addressed my point satisfactorily. The new data they obtained have provided strong support to the dual GnRH model they are proposing.

Reviewer #2 (Remarks to the Author):

The idea that CCK can regulate FSH is not new, see: L. Hollander-Cohen, et al, . Int J Mol Sci (2021). " cckr had enormous expression in FSH cellssuggesting a direct link between the gastrointestinal hormone CCK and FSH activity. In fish, CCK seems to act directly on the pituitary."

The literature has already proposed that CCK acts to regulate FSH in the pituitary, furthermore the literature has already proposed and shown that CCK receptors are expressed in the pituitary.

Cholecystokinin is expressed in the hypophysiotropic neurons

The manuscript is confusing in the history of CCK, especially for someone who is not a "peptide" person:

The receptor is cckbr1, is there a cckar? Furthermore if there is cckbr1, is there a cckbr2? CCKA and CCKB ligands, interact with different receptors?

PAGE 5: There needs to be a more careful presentation on the history of the ligands and the receptors including references for information such as " medaka have two CCK paralogs: cholecystokinin a (ccka) and 164 cholecystokinin b (cckb), as well as gastrin (gast) 30: Why are the authors citing the handbook of hormone for this reference on two forms of CCK? Sekiguchi, T. in Handbook of Hormones (eds Yoshio Takei, Hironori Ando, & Kazuyoshi Tsutsui) 680 177-e120B-173 (Academic Press, 2016). This reviewer finds no evidence in the genome data that there are two ligands for MEDAKA cck cholecystokinin [*Oryzias latipes* (Japanese medaka)]
Gene ID: 101165324, updated on 9-Nov-2023
ccka cholecystokinin a [*Danio rerio* (zebrafish)]
Gene ID: 100007763, updated on 8-Mar-2024
cckb cholecystokinin b [*Danio rerio* (zebrafish)]
Gene ID: 100000095, updated on 8-Mar-2024

Do not understand Fig. 3a. Hypophysiotropic neurons expressing ccka and cckb exist in the 838 hypothalamus. 839 a Luciferase assay of cholecystokinin family peptides CCK8 and Gastrin8 for cAMP, 840 Ca²⁺, and MAPK pathways in HeLa cells expressing Cckbr1.

Authors need to change nomenclature: CCK8 is a test for cell viability, it generates a colour, used in cell culture, thus please clearly state that CCK8 is post-translationally cleaved and the FISH form of CCK peptide is: [(D-Y[SO₃H]-L-G-W-M-D-F-NH₂)

How do authors know that: "Note that both ccka and cckb genes result in the identical deduced 8-amino acid residue 168 peptide CCK8 " when the information on ccka and cckb in Medaka fish is lacking ? Again need references for this information.

Is cck really one gene with a polymorphism related to genetic background?? If there is no compelling evidence that there are two genes, then why are they named as such.

Zebrafish appear to have 2 cck genes, are the a.a. sequences the same?

In the summary figure the authors now just write "CCK" but what about the supposed different forms (CCKA and CCKB)?

Authors should not use CCK8 for experiments in Figure 4 until they clarify the CCKA and CCKB: they

need to look at whether there is a different in response between CCKA and CCKB , then jump to this (assumption) that CCKA and CCKB do the same thing (or better explain ccka and cckb)

Without a better clarification of the CCKA and CCKB ligands, it is difficult to understand why the KO of one has no effect and both have a dramatic effect. This concept is not integrated into the finally summary figure.

For the crossing of the mutant animals authors need to show what the oocytes look like as there may be a defects.

Blue sentences are reviewer's comments, and following sentences are our response. Changes made in the main text are highlighted in yellow.

Reviewer #1 (Remarks to the Author):

The authors have addressed my point satisfactorily. The new data they obtained have provided strong support to the dual GnRH model they are proposing.

We thank the reviewer for appreciating our revision. The reviewer's comments have greatly improved the manuscript.

Reviewer #2 (Remarks to the Author):

The idea that CCK can regulate FSH is not new, see: L. Hollander-Cohen, et al, . Int J Mol Sci (2021). " cckr had enormous expression in FSH cellssuggesting a direct link between the gastrointestinal hormone CCK and FSH activity. In fish, CCK seems to act directly on the pituitary."

The literature has already proposed that CCK acts to regulate FSH in the pituitary, furthermore the literature has already proposed and shown that CCK receptors are expressed in the pituitary.

Hollander-Cohen et al. (2021) explored the genes expressed in gonadotrophs using cell-specific RNAseq and found that *cckr* is one of the genes expressed in *fshb*-expressing cells, as pointed out by the reviewer. Because this finding is relevant to our study, we have already cited Hollander-Cohen et al. (2021) in the Discussion section as follows (Line 289-291):

"A cell type-specific RNA-seq study indicated that FSH-expressing cells showed high expression of CCK receptors in another teleost, tilapia ^{43,44}."

From the evidence of RNA-seq analysis, Hollander-Cohen et al. proposed a link between metabolic balance and reproductive status. However, it did not mention the possibility of hypothalamic CCK regulating FSH release. Thus, we considered that the information provided by Hollander-Cohen et al. related to our study has been sufficiently described in the manuscript.

Cholecystokinin is expressed in the hypophysiotropic neurons

The manuscript is confusing in the history of CCK, especially for someone who is not a "peptide" person:

The receptor is *cckbr1*, is there a *cckar*? Furthermore if there is *cckbr1*, is there a *cckbr2*?

CCKA and CCKB ligands, interact with different receptors?

In response to this comment, we revisited the genome database, such as NCBI or Ensembl, to confirm the nomenclature of CCK and CCK receptors. According to their information, teleosts have two paralogous CCK proteins, *Ccka* and *Cckb*, and three CCK receptors, *Cck1r*, *Cck2ra*, and *Cck2rb* (*Cck2rb* is synonymous with *Cckbr1* in the previous manuscript). By comparing the amino acid

sequences, we confirmed that the sequences of CCK-8s (the post-translationally cleaved and sulfated forms of 8-amino acid peptides of CCK) derived from *ccka* and *cckb* genes are identical. Because Cck-8s have binding capacity for all three receptor types, both Ccka and Cckb have binding capacity for all three receptors. In addition, Gastrin-8s (the post-translationally cleaved and sulfated forms of 8-amino acid peptides of Gastrin) also have binding capacity for all three receptor types. To clearly indicate the sulfated form, we modified the term “CCK8” to “Cck-8s” in the manuscript (Matsuda et al., 2020).

We agree that Ccka/Cckb and Cckar/Cckbr is a misleading nomenclature because both Ccka and Cckb interact with both Cckar and Cckbr. Therefore, we decided to recruit another nomenclature using numbers instead of the alphabet in receptor; Cck1r and Cck2r (Purohit et al., 2012; Zeng et al., 2020).

To clearly present the nomenclature and paralogous relationships of CCK and CCK receptors in the manuscript, we have added the results of phylogenetic tree analysis and the alignment of CCK and CCK receptors. These additional data made it easier for readers to understand the phylogenetic relationships of each ligand and receptor subtypes.

Line 87-88: “Among the metabolic receptors expressed, we found that *cck2rb* had the highest expression (Supplementary Table 1, a phylogenetic tree of Cck receptors is shown in Supplementary Fig. 2).”

Supplementary Fig. 2. Phylogenetic tree of cholecystokinin (cck1 and cck2) receptors.

Maximum likelihood tree of the CCK receptors based on a part of deduced amino acid sequence of each gene. Species name and the accession number for each species are indicated in the tree.

Line 164-166: “Gastrin/Cholecystokinin family; medaka have two CCK paralogs: cholecystokinin a (*ccka*) and cholecystokinin b (*cckb*), as well as gastrin (*gast*)³⁰ (Phylogenetic tree, Supplementary Fig. 6; sequence alignment, Supplementary Fig. 7).”

Supplementary Fig. 6. Phylogenetic tree of medaka cholecystokinin (*cckA* and *cckB*) and gastrin (*gast*).

Maximum likelihood tree of the *cckA*, *cckB*, and *gast* based on deduced amino acid sequence. Species name and the accession number for each species are indicated in the tree.

Supplementary Fig. 7. Alignment of amino acid sequence of vertebrate CCK and gastrin.

Alignment of medaka (MedakaCcka, ENSORLT000000488.1; MedakaCckb, ENSORLT0000007035.1; *gast*, ENSORLT0000015711.1) compared with zebrafish (ZebraCcka, ENSDART00000134202.1; ZebraCckb, ENSDART00000163899.1; ZebraGast, XP_021335429.1), gar (GarCck, ENSLOCT00000001503.1; GarGast, XP_006638439.1) and chicken (ChickCck, ENSGALT00000057710.2; ChickGast, ENSGALT00000043859.2). The red horizontal line indicates the C-terminal CCK octapeptide (CCK-8). Black shading represents identical residues while gray shading represents similar residues.

We summarized the changes in nomenclature here.

“CCK8” has been changed to “CCK-8s”.

“Gastrin-8” has been changed to “Gastrin-8s”.

“Cckar” has been changed to “Cck1r”.

“Cckbr1” has been changed to “Cck2rb”.

“Cckbr2” has been changed to “Cck2ra”.

PAGE 5: There needs to be a more careful presentation on the history of the ligands and the receptors including references for information such as “medaka have two CCK paralogs: cholecystokinin a (*ccka*) and 164 cholecystokinin b (*cckb*), as well as gastrin (*gast*) 30: Why are the authors citing the handbook of hormone for this reference on two forms of CCK? Sekiguchi, T. in Handbook of Hormones (eds Yoshio Takei, Hironori Ando, & Kazuyoshi Tsutsui) 680 177-e120B-173 (Academic Press, 2016).

This reviewer finds no evidence in the genome data that there are two ligands for MEDAKA

cck cholecystokinin [*Oryzias latipes* (Japanese medaka)]

Gene ID: 101165324, updated on 9-Nov-2023

ccka cholecystokinin a [*Danio rerio* (zebrafish)]

Gene ID: 100007763, updated on 8-Mar-2024

cckb cholecystokinin b [*Danio rerio* (zebrafish)]

Gene ID: 100000095, updated on 8-Mar-2024

This may be a misunderstanding by the reviewer. Medaka has two cholecystokinin genes. The reviewer likely searched in NCBI with the word "cck" and only found one result in medaka. However, when we searched with the word "cholecystokinin", we found two cholecystokinin genes in medaka, which are *ccka* and *cckb* as mentioned in our manuscript (see Gene ID: 101165324 and Gene ID: 101167231). These genes are located on chromosomes 16 and 11, respectively. In Ensembl, we also found two distinct *cck* genes, *ccka* and *cckb* (ENSORLG00000005949 and ENSORLG00000005594). Therefore, there is no doubt that medaka possesses *ccka* and *cckb*, which were analyzed in the present study.

To address the reviewer's comment and prevent any further misunderstanding, we have included a phylogenetic tree as a supplementary figure and referenced it in the main text (Please refer to the above comment). Additionally, we have added the following reference, which demonstrated synteny of *ccka* and *cckb* genes to clearly show that they are two paralogous genes in medaka.

Dupré, D., Tostivint, H. Evolution of the gastrin–cholecystokinin gene family revealed by synteny analysis. *Gen Comp Endocrinol* **195**, 164–173 (2014).

Also, we added a discussion regarding the situation of co-expression of paralogous genes to emphasize that *ccka* and *cckb* are co-expressing (Line 343-346).

“Similar to the co-expression of *ccka* and *cckb*, paralogous genes sometimes remain co-localized after gene duplication for a relatively long evolutionary period^{68,69}. Further elucidation of the meaning of this co-expression of *ccka* and *cckb* could be an intriguing topic for a future research.”

Do not understand Fig. 3a. Hypophysiotropic neurons expressing *ccka* and *cckb* exist in the 838 hypothalamus. 839 a Luciferase assay of cholecystokinin family peptides CCK8 and Gastrin8 for cAMP, 840 Ca²⁺, and MAPK pathways in HeLa cells expressing *Cckbr1*.

Authors need to change nomenclature: CCK8 is a test for cell viability, it generates a colour, used in cell culture, thus please clearly state that CCK8 is post-translationally cleaved and the FISH form of CCK peptide is: [(D-Y[SO₃H]-L-G-W-M-D-F-NH₂)

In response to the comment, we have added a more detailed explanation about CCK-8s in the manuscript as follows (Line 166-169):

“Because the post-translationally cleaved and sulfated forms of 8-amino acid peptide of *Ccka* and *Cckb* (CCK-8s: DY(SO₃H)LGWMDF-NH₂ in medaka) and Gastrin (Gastrin-8s: DY(SO₃H)RGWLDF-NH₂ in medaka) are reported to show sufficient biological activity, we used them for the luciferase reporter assay”.

How do authors know that: “Note that both *ccka* and *cckb* genes result in the identical deduced 8-amino acid residue 168 peptide CCK8 “ when the information on *ccka* and *cckb* in Medaka fish is lacking ? Again need references for this information.

Is *cck* really one gene with a polymorphism related to genetic background?? If there is no compelling evidence that there are two genes, then why are they named as such.

This question is based on the reviewer's misunderstanding that there is no proof that medaka have two *cck* genes. As mentioned in the previous comment, the gene and amino acid sequences of the two *cck* genes are available in databases such as NCBI and Ensembl. By comparing the amino acid sequences, we have confirmed that the CCK8 sequences derived from *ccka* and *cckb* genes are identical. Additionally, phylogenetic tree analysis led to the naming of *ccka* and *cckb*, following the nomenclature used for other fish species. For better understanding, we have added a supplementary figure that demonstrates the amino acid sequences of the Gastrin/Cholecystokinin family, including medaka *ccka* and *cckb*, as follows:

Line 164-166: “Gastrin/Cholecystokinin family; medaka have two CCK paralogs: cholecystokinin a (*ccka*) and cholecystokinin b (*cckb*), as well as gastrin (*gast*)³⁰ (Phylogenetic tree, Supplementary Fig. 6; sequence alignment, Supplementary Fig. 7).”

Zebrafish appear to have 2 *cck* genes, are the a.a. sequences the same?

As the review pointed out, zebrafish have two *cck* genes. In zebrafish, unlike medaka, there is one amino acid substitution in the deduced amino acid sequence of CCK-8 (Supplementary Fig. 7). The detailed

information is indicated in the supplementary figure that lists the deduced amino acid sequences of CCK in various species (Supplementary Fig. 7).

In the summary figure the authors now just write “CCK” but what about the supposed different forms (CCKA and CCKB)?

Following this comment, we have added “Ccka/Cckb” in the summary figure.

Authors should not use CCK8 for experiments in Figure 4 until they clarify the CCKA and CCKB: they need to look at whether there is a difference in response between CCKA and CCKB, then jump to this (assumption) that CCKA and CCKB do the same thing (or better explain ccka and cckb)

Following this and the above comments, we have revised the manuscript to include a more detailed explanation of CCK-8s before presenting the results shown in Figure 4 (Please refer to the above comment).

Without a better clarification of the CCKA and CCKB ligands, it is difficult to understand why the KO of one has no effect and both have a dramatic effect. This concept is not integrated into the final summary figure.

As explained above, to clarify the differences in sequence between Ccka and Cckb, we have included new supplementary figures, a phylogenetic tree analysis and amino acid sequences in the manuscript (Please see the above comments).

For the crossing of the mutant animals authors need to show what the oocytes look like as there may be a defects.

We have already presented the gross morphology of the ovaries of the *cck2rb* KO (Fig. 1c) and *ccka/cckb* double KO strains (Fig. 5a). The *ccka/cckb* double KO females, as well as the *cck2rb* KO females, have severely degenerated ovaries and are unable to lay eggs, clearly indicating a failure to produce mature eggs. In contrast, females of the other genotypes show no abnormalities in ovarian gross morphology or number of eggs laid, suggesting that they have normal ovaries. In addition, histological analysis of the ovaries has been performed on *cck2rb* KO females, confirming the absence of mature eggs (Fig.1f). Furthermore, all genotypes except for *cck2rb* KO and *ccka/cckb* double KO showed normal fertilization rate (Supplementary Fig.4, Supplementary Fig.14). We consider these data to be sufficient to address that spawned eggs are normal.